# Genome-wide base editor screen identifies regulators of protein abundance in yeast

Olga T Schubert[1,2,3,4,5,6]*, Joshua S Bloom[1,2,3,4], Meru J Sadhu[1,2,3,4†], Leonid Kruglyak[1,2,3,4]*

[1]Department of Human Genetics, University of California, Los Angeles, Los Angeles, United States; [2]Department of Biological Chemistry, University of California, Los Angeles, Los Angeles, United States; [3]Howard Hughes Medical Institute, University of California, Los Angeles, Los Angeles, United States; [4]Institute for Quantitative and Computational Biology, University of California, Los Angeles, Los Angeles, United States; [5]Department of Environmental Systems Science, Swiss Federal Institute of Technology (ETH), Zürich, Switzerland; [6]Department of Environmental Microbiology, Swiss Federal Institute of Aquatic Science and Technology (Eawag), Dübendorf, Switzerland

*For correspondence:
olga.schubert@eawag.ch (OTS);
lkruglyak@mednet.ucla.edu (LK)

Present address: †National Human Genome Research Institute, National Institutes of Health, Bethesda, United States

Competing interest: The authors declare that no competing interests exist.

**Abstract** Proteins are key molecular players in a cell, and their abundance is extensively regulated not just at the level of gene expression but also post-transcriptionally. Here, we describe a genetic screen in yeast that enables systematic characterization of how protein abundance regulation is encoded in the genome. The screen combines a CRISPR/Cas9 base editor to introduce point mutations with fluorescent tagging of endogenous proteins to facilitate a flow-cytometric readout. We first benchmarked base editor performance in yeast with individual gRNAs as well as in positive and negative selection screens. We then examined the effects of 16,452 genetic perturbations on the abundance of eleven proteins representing a variety of cellular functions. We uncovered hundreds of regulatory relationships, including a novel link between the GAPDH isoenzymes Tdh1/2/3 and the Ras/PKA pathway. Many of the identified regulators are specific to one of the eleven proteins, but we also found genes that, upon perturbation, affected the abundance of most of the tested proteins. While the more specific regulators usually act transcriptionally, broad regulators often have roles in protein translation. Overall, our novel screening approach provides unprecedented insights into the components, scale and connectedness of the protein regulatory network.

## Editor's evaluation

This paper describes a novel CRISPR-based screening method that allows probing interactions between a large set of specific mutations and the abundance of specific proteins, and, more generally, investigate the spectrum of effects that (point) mutations can have on protein abundance. This novel technique complements existing strategies for measuring effects of genetic perturbations on transcript levels, which is important as for some genes mRNA and protein levels may not correlate well. The ability to measure proteins directly therefore promises to help close a gap in our understanding of the links between genotype and phenotype, and the strategy is broadly applicable beyond the current study.

## Introduction

CRISPR-based genetic screens using Cas9 or modified versions thereof, such as the CRISPR base editor or CRISPRi, provide an unprecedented opportunity to study the effects of thousands of genetic perturbations on gene expression in a pooled format (*Adamson et al., 2016*; *Cuella-Martin et al., 2021*; *Datlinger et al., 2017*; *Després et al., 2020*; *Dixit et al., 2016*; *Gasperini et al., 2020*; *Hanna et al., 2021*; *Jaitin et al., 2016*). The identity of genetic perturbations with phenotypic effects can be directly determined by sequencing guide RNAs, thereby eliminating the need for genome sequencing to identify causal mutations. To date, such studies have focused on the effects of genetic perturbations on mRNA expression, while the effects on protein expression remain unexplored.

Proteins are gene products that play many key roles in cells, serving as essential structural components, enzymes, and constituents of signaling networks, from receptors to transcription factors. Their abundance is extensively regulated both transcriptionally and post-transcriptionally. For practical reasons, mRNA abundance is often studied as a proxy for protein abundance, but the correspondence between the two is less strong than is usually implicitly assumed (*Buccitelli and Selbach, 2020*; *Liu et al., 2016a*). Therefore, it is important to study regulation of protein abundance directly.

Here, we developed a novel CRISPR base editor screen to study how tens of thousands of individual mutations affect the abundance of selected proteins in yeast. Using this approach, we tested the effects of 16,452 coding mutations on the abundance of eleven yeast proteins selected to represent a variety of molecular and cellular functions. We identified hundreds of novel regulatory relationships, including both specific and broad regulators of protein abundance. The results demonstrate that base editor screens can be used to gain general insights into how protein abundance is encoded in the genome and altered by changes in DNA sequence.

## Results

### Base editing in yeast is efficient and predictable

To introduce genetic perturbations across the yeast genome in a targeted fashion, we used the CRISPR base editor version 3 (BE3) (*Komor et al., 2016*). This system consists of an engineered fusion of Cas9 and a cytidine deaminase that does not introduce DNA double-strand breaks but instead converts cytosine to uracil, thereby effecting a C-to-T substitution at a target site defined by the guide

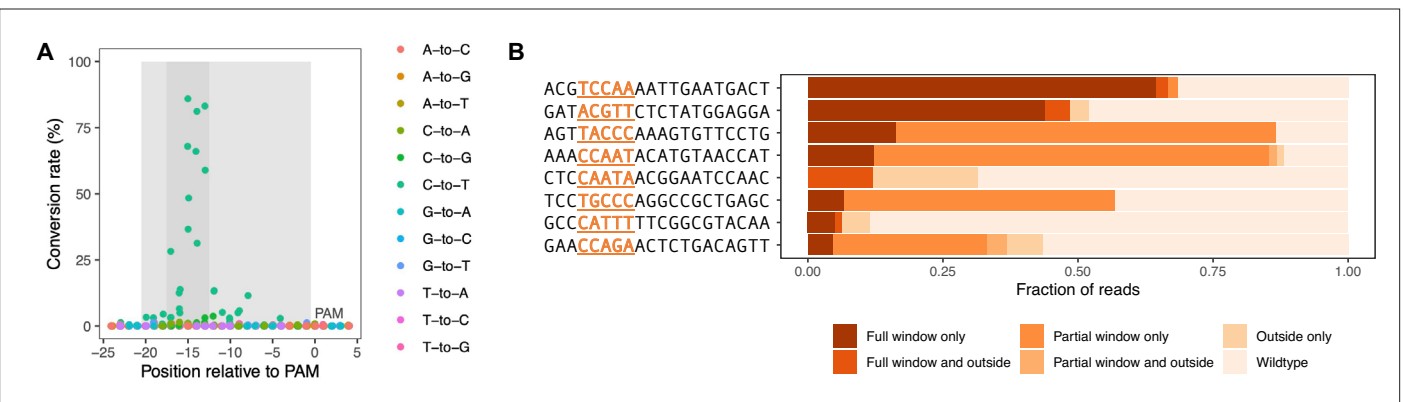

**Figure 1.** Base editor characterization in yeast. For eight gRNAs, following base editing for 44 hr, the genomic target site was PCR-amplified and subjected to deep sequencing. (**A**) Base editing outcome for the eight gRNAs with target genomic regions aligned by PAM site. Light gray shading delineates the gRNA target site, and dark gray shading delineates the window of greatest editing (13–17 base pairs upstream of the PAM sequence). (**B**) Base editing outcome classified by editing pattern for each of the eight gRNAs tested. The editing window is indicated in the gRNA sequence. The horizontal axis of the plot reflects the fraction of reads for each editing pattern out of all reads acquired for the respective gRNA. Only alleles with >1% frequency were classified.

The online version of this article includes the following figure supplement(s) for figure 1:

**Figure supplement 1.** Base editor and gRNA plasmid design for base editing in *S.cerevisiae*.

**Figure supplement 2.** Base editing profile at the genomic target site of each of eight gRNAs.

**Figure supplement 3.** Base editing rate as assessed by editing of the GFP encoding gene.

RNA (gRNA) (*Komor et al., 2016*). To adapt the base editor for use in yeast, we created a plasmid which contains the base editor under a galactose-inducible promoter, as well as a plasmid optimized for cloning of both gRNAs individually and gRNA libraries in bulk (*Figure 1—figure supplement 1*). These plasmids were transformed into yeast cultures, and base editor expression was induced with galactose. First, we assessed base-editing efficiency and the target window in yeast by amplicon-sequencing the genomic target loci of eight individual gRNAs after base editor expression was induced for 44 hr. We found that the target window and mutagenesis pattern are very similar to those described in human cells: 95% of edits are C-to-T transitions, and 89% of these occurred in a five-nucleotide window 13–17 base pairs upstream of the PAM sequence (*Figure 1A*; *Figure 1—figure supplement 2*; *Komor et al., 2016*). Editing efficiency was variable across the eight gRNAs and ranged from 4% to 64% if considering only cases where all Cs in the window are edited; percentages are higher if incomplete edits are considered, too (*Figure 1B*).

To assess base editing efficiency over time, we designed two gRNAs predicted to lead to loss-of-function mutations in GFP. We expressed the base editor and these gRNAs in yeast cells with a genomically integrated GFP gene. Depending on the gRNA, cells lost the GFP signal at different rates; however, for both gRNAs, a majority of cells had lost the GFP signal after 24 hr (*Figure 1—figure supplement 3*).

## Base editing in yeast allows screening in pooled format

To assess base editor performance in a pooled screen format, we used a library of all 90 gRNAs suitable for targeting the *CAN1* gene (*Supplementary file 1*). Loss-of-function mutations in *CAN1* render yeast resistant to the toxic arginine analog canavanine. We exposed base-edited yeast cultures to canavanine for 24 and 48 hr and observed a strong enrichment of reads from gRNAs predicted to introduce premature stop codons ($p < 2.2e-16$, $\chi^2 = 1737$, df=1, chi-squared test), as well as a

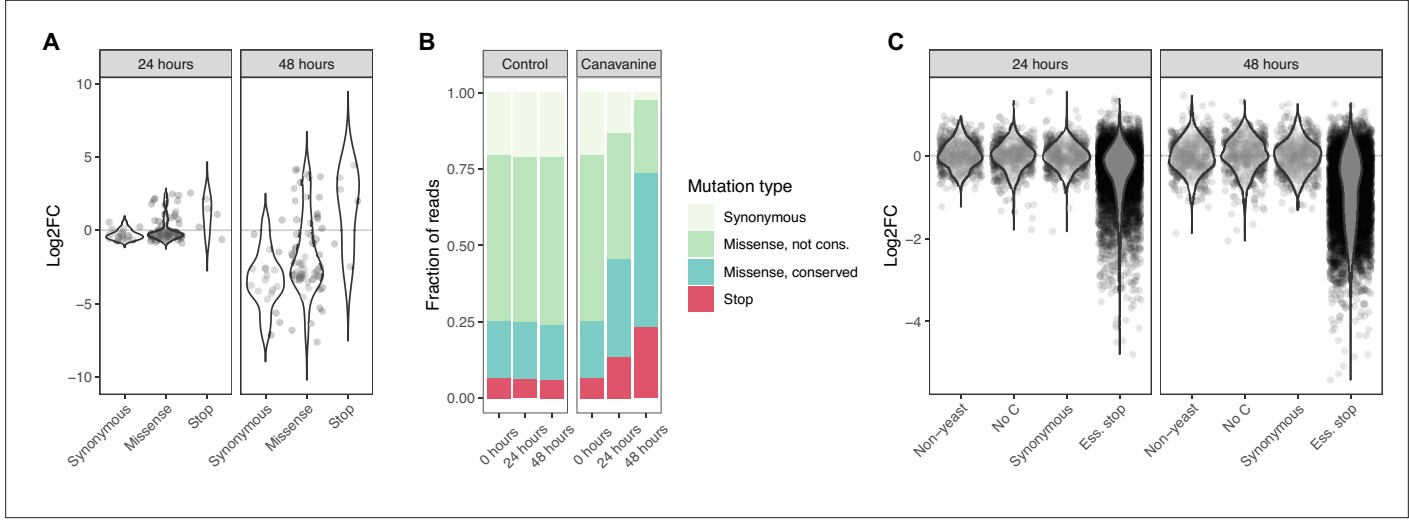

**Figure 2.** Base editor screens identify loss-of-function mutations in positive and negative selection schemes. (**A**) For the canavanine screen, canavanine was added to cultures of cells that were base edited with a library of 90 gRNAs targeting the *CAN1* gene. The log$_2$ fold change of reads per gRNA at 24 and 48 hr after canavanine treatment compared to before treatment is shown. gRNAs are grouped by the type of mutation they are predicted to introduce. (**B**) The fraction of reads that map to gRNAs introducing different types of mutations into *CAN1* 48 hr after starting the canavanine treatment is shown. gRNAs expected to introduce missense mutations are split according to the conservation of the expected amino acid mutation, where 'conserved' is defined by PROVEAN scores <–5 and 'not conserved' by PROVEAN scores ≥–5. (**C**) For the fitness screen, yeast cultures were base edited using a library of gRNAs predicted to introduce premature stop codons into essential genes. These mutations are therefore expected to lead to a drop out of affected cells (and the responsible gRNA) from the cultures. The log$_2$ fold change of reads per gRNA 24 and 48 hr compared to 0 hr after induction of base editing is shown. The gRNAs belong to either of four classes: (**i**) not targeting the yeast genome, (**ii**) no cytosine residue in the base editing window, (**iii**) predicted to introduce synonymous mutations only, (**iv**) predicted to introduce stop codons into essential genes.

The online version of this article includes the following source data and figure supplement(s) for figure 2:

**Source data 1.** Results of canavanine survival screen with 90 gRNAs targeting the *CAN1* gene.

**Source data 2.** Results of fitness screen with 5430 gRNAs predicted to introduce stop codons into essential genes.

**Figure supplement 1.** Effect of sequence context on base editing efficiency.

depletion of reads from gRNAs predicted to introduce synonymous mutations in *CAN1* (p<2.2e-16, $\chi^2$=1637, df=1, chi-squared test) (*Figure 2A and B*; *Figure 2—source data 1*). Eleven of the 65 gRNAs predicted to introduce missense mutations resulted in an enrichment similar to that of the stop codons. The amino acid substitutions introduced by these eleven gRNAs therefore likely disrupt the function of *CAN1*. Most of these missense mutations with a strong effect are at highly conserved amino acid residues, as reflected by their more negative PROVEAN scores (*Choi et al., 2012*); seven of the eleven amino acid substitutions have a PROVEAN score <–5, indicating high conservation (p=0.007, OR=0.15, Fisher's exact test) (*Figure 2B*).

To further characterize base editing efficiency in yeast in a pooled screen format, we designed a library containing all 5430 gRNAs predicted to introduce stop codons into essential genes, together with three sets of 500 control gRNAs each (random gRNA sequences not targeting the yeast genome, gRNAs without a C in the targeting region, gRNAs introducing synonymous mutations) (*Supplementary file 1*). We evaluated the effects of these gRNAs on yeast survival over 48 hr. As expected, cells transformed with control gRNAs remained in the culture at stable levels, whereas cells transformed with gRNAs introducing stop codons into essential genes tended to drop out over time (59% displayed $\log_2$ fold change <–0.5 and false-discovery rate (FDR) for depletion <0.05) (*Figure 2C*; *Figure 2—source data 2*). Despite the observed heterogeneity of editing outcomes across individual gRNAs (*Figure 1B*), in these pooled positive and negative selection screens we see clear signals for gRNAs predicted to introduce stop codons into the *CAN1* gene and into essential genes (effect for 67% and 59% of gRNAs, respectively).

We next asked whether DNA sequence context in the targeted region could explain which gRNAs introduced stop codons with high efficiency. To address this, we built a lasso regression model with over 8000 sequence features per gRNA and found that it could explain 37% of the observed variance in gRNA depletion in the screen of essential genes (Pearson's $R^2$=0.37, p<2.2e-16) (*Figure 2—figure supplement 1*). In particular, the base preceding the target C has a strong effect—a T or another C produces much higher editing rates than a G. These findings in yeast are in line with recent reports in human cell lines and can help guide the design of future gRNA libraries (*Arbab et al., 2020*; *Song et al., 2020*).

In summary, we developed a set of experimental resources for pooled base editor screens in yeast, including a plasmid system suitable for large-scale gRNA library cloning. We also demonstrated that these resources can be used to carry out base editor screens that are able to identify functionally important mutations in yeast using both positive selection (canavanine resistance screen) and negative selection (fitness screen).

## A base editor screen to identify genetic effects on protein abundances

We next set out to develop a genome-wide base editor screen that would allow us to study genetic loci that affect the abundance of selected proteins. To this end, we compiled a library of 16,452 gRNAs predicted to introduce three classes of mutations: stop codons in nonessential genes, as well as missense mutations at highly conserved positions in essential genes and in nonessential genes (Methods and *Supplementary file 1*). We expect most genetic perturbations introduced by these gRNAs to have deleterious effects on the target genes. We introduced this gRNA library and the galactose-inducible base editor into yeast strains in which one protein of interest is chromosomally tagged with green fluorescent protein (GFP) (*Howson et al., 2005*; *Huh et al., 2003*). The GFP fluorescence level in these strains reports the abundance of the tagged protein (*Newman et al., 2006*). We selected eleven strains with GFP-tagged proteins that cover a range of cellular functions. The target proteins include the Hsp70 chaperone Ssa1, which is involved in protein folding, nuclear transport and ubiquitin-dependent degradation of short-lived proteins and is a marker for heat shock; as well as the flavohemoglobin Yhb1, which is involved in oxidative and nitrosative stress responses and is a marker for oxidative stress (*Soste et al., 2014*). Furthermore, we included the ribonucleotide reductase subunit Rnr2, which is involved in dNTP synthesis, the ribosomal protein Rpl9A, the histone Htb2, and several proteins involved in metabolism, including the fatty acid synthases Fas1 and Fas2, the enolase Eno2, which acts in glycolysis and gluconeogenesis, and the three glyceraldehyde-3-phosphate dehydrogenase (GAPDH) isoenzymes Tdh1, Tdh2, and Tdh3. Tdh3 is among the most highly expressed proteins in yeast and the main GAPDH enzyme, whereas the exact functional roles and regulation of Tdh1 and Tdh2 are less well understood.

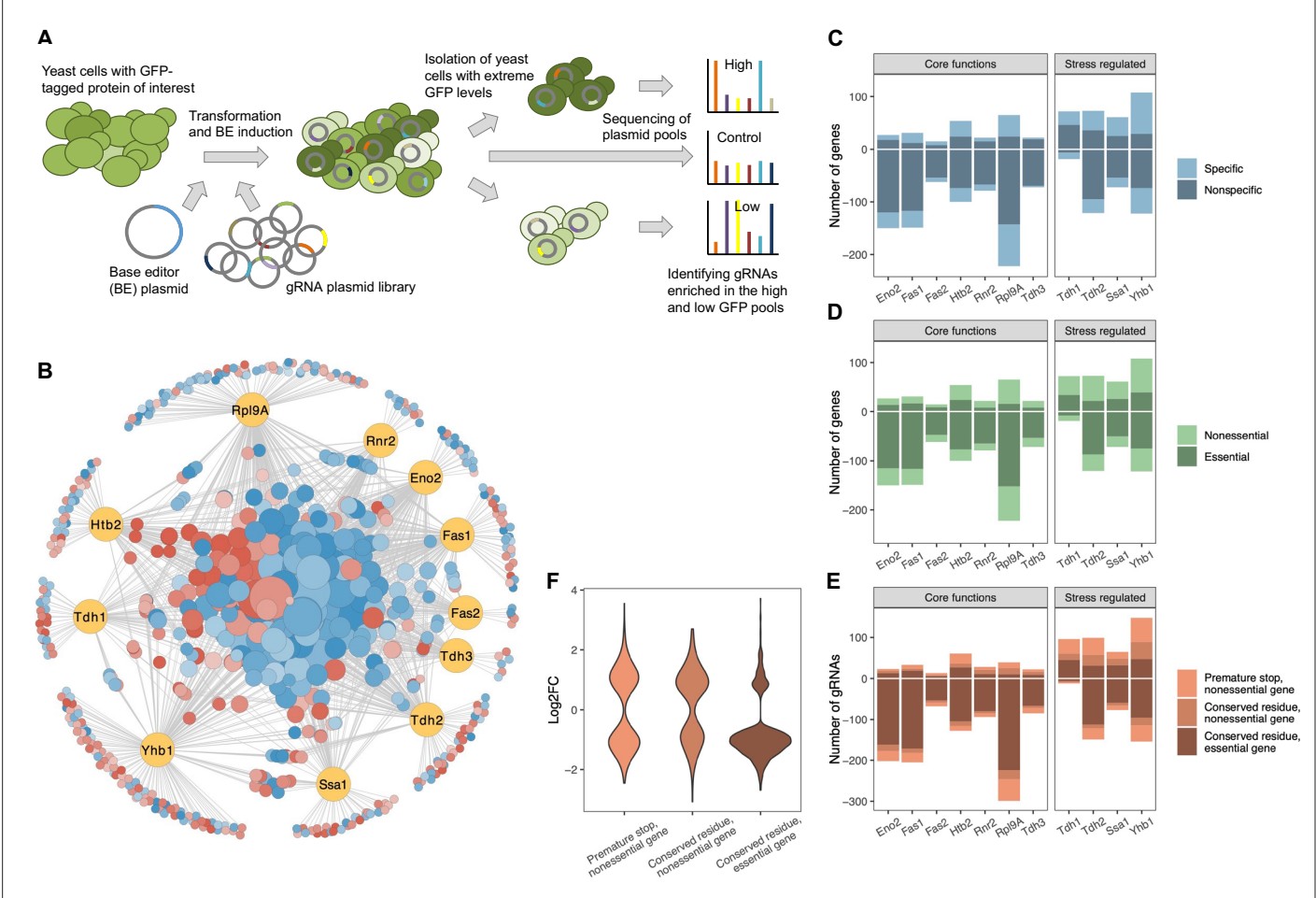

**Figure 3.** A CRISPR base editor screen for protein abundance. (**A**) Schematic overview of the screen. (**B**) Network representing all identified gene-protein relationships. In yellow are the eleven proteins and in blue-red are all the gene perturbations that affect at least one of the proteins significantly (FDR<0.05). On the outer circle are gene perturbations that affect only a single protein, whereas on the inside are those that affect two or more proteins. Node sizes correlate with the number of proteins affected. Colors indicate whether a perturbation (predominantly) increases (red) or decreases (blue) the protein(s). The figure was created with Cytoscape (**_Shannon et al., 2003_**). (**C**) For each protein, the number of gene perturbations that cause a significant increase (positive vertical axis) or decrease (negative vertical axis) (FDR<0.05). The darker shade indicates gene perturbations that affect only one or two of the eleven proteins ('specific'), whereas the lighter shade indicates gene perturbations that affect three or more of the eleven proteins ('nonspecific'). (**D**) Same as in (C) but the darker shade indicates perturbations of essential genes, whereas the lighter shade indicates perturbations of nonessential genes. (**E**) Same as in (C) but the different shades reflect different types of expected mutations introduced by the particular gRNA. (**F**) Effect sizes (log$_2$ fold changes) of gRNAs that cause a significant change in protein abundance grouped by the type of expected mutation introduced by each gRNA (FDR<0.05).

The online version of this article includes the following source data and figure supplement(s) for figure 3:

**Source data 1.** Results of protein abundance screens with 16,452 gRNAs.

**Figure supplement 1.** Correlation between replicates and among gRNAs targeting the same mutation.

**Figure supplement 2.** Correlation between the effect of genetic perturbations on mRNA and protein abundances.

**Figure supplement 3.** Correlation between the number of gene perturbations significantly affecting a protein and the absolute abundance of that protein.

**Figure supplement 4.** Correlation between number and effect size of perturbations with significant effects and their target site across the gene.

After performing base editing in each of the eleven GFP strains, we isolated cells with very high and very low GFP intensity by fluorescence-activated cell sorting (**_Figure 3A_**). We next sequenced gRNA plasmids extracted from the sorted cells; the gRNAs enriched or depleted in these populations correspond to genetic perturbations that increase or decrease abundance of the protein of interest (**_Figure 3A_**). We identified 1020 gRNAs with a significant effect on the abundance of at least one of

the eleven proteins (FDR<0.05) (*Figure 3—source data 1*). Effects of gRNAs in replicate experiments showed significant correlation (Pearson's $R^2$=0.37, p<2.2e-16) (*Figure 3—figure supplement 1A*). To further assess the reproducibility of the screen, we included gRNA pairs that were predicted to introduce the same mutation into the genome. For 78 of these gRNA pairs, at least one gRNA had a significant effect (FDR<0.05) on at least one of the eleven proteins; their effects were well correlated (Pearson's $R^2$=0.43, p<2.2e-16) (*Figure 3—figure supplement 1B*). For the 20 gRNA pairs in which both gRNAs had a significant effect, the correlation was even higher (Pearson's $R^2$=0.81, p=8.8e-13) (*Figure 3—figure supplement 1C*). These findings show that the significant gRNA effects that we identify have a low false positive rate, but they also suggest that many real gRNA effects are not detected in the screen due to limitations in statistical power. The effects of genetic perturbations on protein abundances correlate well with the effects on mRNA abundances measured in an earlier study (*Kemmeren et al., 2014*) when considering only effects that are significant in both studies (Pearson's $R^2$=0.62, p<5.3e-15) (*Figure 3—figure supplement 2*).

When multiple gRNAs that introduce different mutations into the same gene had a significant effect on a protein's expression, their direction of effect was concordant in 98% of the cases. We therefore combined gRNA effects into a gene-level effect for all subsequent analyses, unless noted otherwise. Genetic perturbation of 710 genes had a significant effect on the abundance of at least one of the eleven proteins (FDR<0.05) (*Figure 3—source data 1*). Some of these genes may be direct regulators of the tested proteins, while others may affect protein abundances more indirectly. For simplicity and for lack of a better term, we will, in the following, refer to these genes as regulators without implying a direct mechanistic link.

## Extensive genetic network underlies protein abundance regulation

The effects of genetic perturbations in 710 genes on the abundance of one or more of the eleven target proteins reveal an extensive network of regulatory relationships (*Figure 3B*). On average, the abundance of a protein was significantly affected by 156 gene perturbations (*Figure 3C*). Of these gene-protein relationships, an average of 53 (34%) are specific (defined here as gene perturbations that affect only one or two of the eleven target proteins) whereas 103 (66%) are nonspecific (the gene perturbation affects three or more of the eleven proteins). Furthermore, we found that, on average, protein abundance decreases more frequently than it increases in response to the gene perturbations tested (106 vs 50, p=8.6e-6, binomial test) (*Figure 3C*). However, this trend is strongly dependent on the functional role of the protein. Abundance of proteins involved in core cellular functions mostly decreases, whereas abundance of stress-related proteins is just as likely to increase as it is to decrease (p=2.9e-12 and p=0.8, respectively, binomial test) (*Figure 3C*); this trend is independent of the absolute abundance of a protein (*Figure 3—figure supplement 3*).

The predominance of decreases in protein abundance in response to gene perturbations is driven by perturbations in essential genes, which on average decrease protein abundance in 80% of cases, whereas perturbations in nonessential genes do so in only 49% of cases (*Figure 3D*). Furthermore, perturbations in essential genes are generally more likely to significantly alter protein abundance than are perturbations in nonessential genes (p<2.2e-16, $\chi^2$=384, df=1, chi-squared test) (*Figure 3D and E*). For nonessential genes, premature stop codons are more likely to alter protein abundance than are missense mutations (p=1.2e-9, $\chi^2$=37, df=1, chi-squared test) (*Figure 3E*), and the average effect size of changes caused by stop codons is larger (p=0.0028, t=–3.0, df=496, two-sided t-test) (*Figure 3F*). We also find that mutations towards the end of a gene tend to have fewer and smaller effects, in line with previous observations (*Figure 3—figure supplement 4*; *Sadhu et al., 2018*).

## Base editor screen identifies known and novel regulatory relationships

The hundreds of regulatory relationships revealed by the base editor screen include many previously known regulators of the proteins under study (*Figure 4A*). For example, proteins involved in glycolysis (Eno2 and Tdh1/2/3) are strongly induced upon perturbation of other genes involved in glycolysis. Perturbation of Gcr1, a key transcription factor for glycolytic genes, leads to reduced Eno2 abundance, consistent with its role as a transcriptional activator (*Baker, 1986*; *Holland et al., 1987*). Similarly, histone Htb2 abundance is strongly reduced upon perturbation of known regulators of histone gene transcription (Hir3, Spt10, Spt21) (*Kurat et al., 2014*). Ribosomal protein Rpl9A abundance is also strongly reduced upon perturbation of other ribosomal genes, as well as of genes involved in

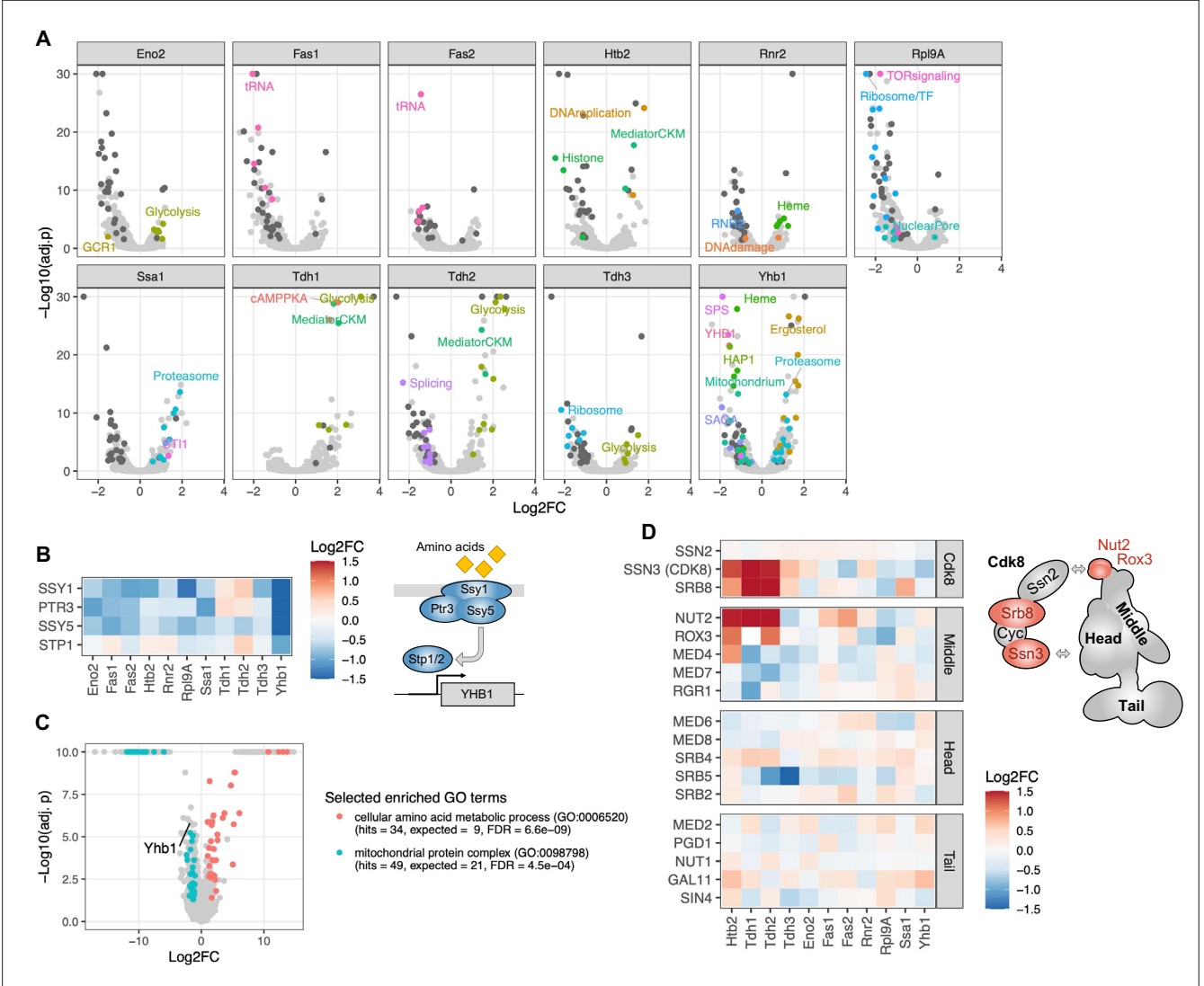

**Figure 4.** Regulators of protein abundance. (**A**) Each volcano plot contains the results for a single protein, with each dot representing the effect of a gene's perturbation on that protein. The vertical axis represents the negative logarithm of the FDR-corrected p-value. Selected dots are colored to highlight functional classes. Dark gray dots indicate genes that significantly affect eight or more proteins (FDR<0.05). (**B**) Heatmap showing effects of gene perturbations in the SPS amino acid sensor on the eleven proteins. In the scheme, all subunits with significant effect on Yhb1 are colored (FDR<0.05). (**C**) Volcano plot showing abundance changes of proteins in the *SSY5* G638Q mutant compared to wild type as measured by LC-MS proteomics. The vertical axis represents the negative logarithm of the FDR-corrected p-value. Selected enriched functional categories (gene ontology (GO) terms) are colored. (**D**) Heatmap showing effects of gene perturbations in Mediator on the eleven proteins. In the scheme, all subunits with significant effects on Htb2 are colored (FDR<0.05); arrows indicate suggested interaction points between the Cdk8 module and mediator (*Tsai et al., 2014*; *Tsai et al., 2013*). The color scale for the heatmaps is capped at –1.5 and 1.5.

The online version of this article includes the following source data for figure 4:

**Source data 1.** Results of mass spectrometry-based proteomics analysis of individual mutant strains.

TOR signaling (Tor2, Asa1, Kog1, Rtc1, Sch9), including the TOR-regulated activators of ribosomal gene transcription, Fhl1 and Ifh1, which are known to act together with the broad transcription factor Rap1 to induce expression of ribosomal genes (*Rudra et al., 2005*).

We also find numerous novel regulatory relationships. For example, the abundance of the stress-responsive flavohemoglobin/oxidoreductase Yhb1 is reduced by perturbation of one of its prime transcriptional activators, heme activator protein Hap1, as well as perturbations of a heme transporter and six genes involved in heme biosynthesis (*Cassanova et al., 2005*). This is in accordance with Yhb1 being a heme protein itself (*Zhu and Riggs, 1992*). A novel regulator of Yhb1 identified

in our screen is the three-subunit Ssy1-Ptr3-Ssy5 (SPS) amino acid sensor. We find that perturbation of each of the three subunits, as well as of one of its two downstream effectors, the transcription factor Stp1, results in reduced Yhb1 abundance (*Figure 4B*). We independently confirmed the new regulatory relationship between Yhb1 and SPS by liquid-chromatography-coupled tandem mass spectrometry (LC-MS) proteomics analysis of one of the SPS mutants identified in our screen, *SSY5* G638Q (*Figure 4C* and *Figure 4—source data 1*). Furthermore, gene ontology (GO) enrichment analysis of differentially expressed proteins in this mutant revealed that SPS perturbation leads not only to reduced Yhb1 abundance but also to reduced abundance of other mitochondrial proteins, including components of the mitochondrial ribosome, oxidative phosphorylation, and ATP metabolism (FDR=4.5e-4, *Figure 4C*). In contrast, amino acid metabolic processes are broadly induced in the mutant (FDR=6.6e-9, *Figure 4C*), suggesting that SPS signaling not just activates expression of amino acid transporters but also represses amino acid biosynthesis (*Forsberg et al., 2001*). Recently, the presence of extracellular amino acids has been found to induce Yhb1 expression in the distantly related yeast *Candida albicans* (*Danhof et al., 2016*; *Williams and Lorenz, 2020*). Our observations extend this link between amino acids and increased nitrosative and oxidative stress resistance to *S. cerevisiae* and suggest that it is mediated by the SPS amino acid sensor. We propose that SPS senses the presence of external amino acids and activates the transcription factor Stp1, which in turn induces Yhb1 expression, resulting in increased nitrosative and oxidative stress resistance (*Figure 4B*).

In addition to the relatively specific regulators described above, we identified two key complexes that modulate the transcriptional machinery, mediator and SAGA, as regulators of protein abundance. We find particularly large effects on three proteins (Htb2, Tdh1, and Tdh2) upon perturbation of two subunits of the Cdk8 kinase module of mediator, Ssn3 and Srb8 (*Figure 4D*). Moreover, we see large effects on the same proteins for perturbations of the two subunits forming the hook structure on mediator, Nut2 and Rox3, which supports recent reports that this domain is critical for the interaction between the core mediator complex and the Cdk8 kinase module (*Figure 4D*; *Tsai et al., 2014*; *Tsai et al., 2013*). The observation that perturbations of the Cdk8 module increase protein abundance is in accordance with the suggested repressor role of this transiently associating regulatory module of mediator (*Elmlund et al., 2006*). We also find that classical post-transcriptional regulators contribute strongly to protein abundance. For example, Ssa1 and Yhb1 are significantly induced upon perturbation of a number of genes involved in proteasomal degradation (*Figure 4A*).

## Gene perturbations with broad effects on protein abundance often affect protein biosynthetic processes

About half of the 710 gene perturbations with significant effects on protein abundance affect at least two proteins, and some of these alter the abundance of a large fraction of the eleven proteins studied (*Figure 3B* and *Figure 5A*). For instance, perturbations in 29 genes significantly alter the abundance of eight or more proteins (*Figure 5B*, left panel). Of these 29 genes, 21 (72%) have roles in protein translation—more specifically, in ribosome biogenesis and tRNA metabolism (FDR<8.0e-4, *Figure 5C*). In contrast, perturbations that affect the abundance of only one or two of the eleven proteins mostly occur in genes with roles in transcription (e.g., GO:0006351, FDR<1.3e-5). Protein biosynthesis entails both transcription and translation, and these results suggest that perturbations of translational machinery alter protein abundance broadly, while perturbations of transcriptional machinery can tune the abundance of individual proteins. Thus, genes with post-transcriptional functions are more likely to appear as hubs in protein regulatory networks, whereas genes with transcriptional functions are likely to show fewer connections.

One of the broadest regulators is *POP1*, an essential gene involved in ribosomal RNA (rRNA) and transfer RNA (tRNA) maturation (*Figure 5C*). We used LC-MS proteomics to independently confirm that missense mutation H642Y in *POP1* leads to reduced abundance of most of the eleven proteins under study (*Figure 5D* and *Figure 4—source data 1*). Furthermore, the proteomics data revealed that perturbation of *POP1* reduces the abundance of many proteins in the nucleolus (FDR=0.013, *Figure 5—figure supplement 1A*), consistent with the role of *POP1* in rRNA and tRNA processing. We also see the induction of a large number of proteins involved in amino acid biosynthesis (FDR=1.4e-13, *Figure 5—figure supplement 1A*). This observation could represent an indirect response to perturbed ribosome metabolism or a direct role of *POP1* in suppressing amino acid metabolism.

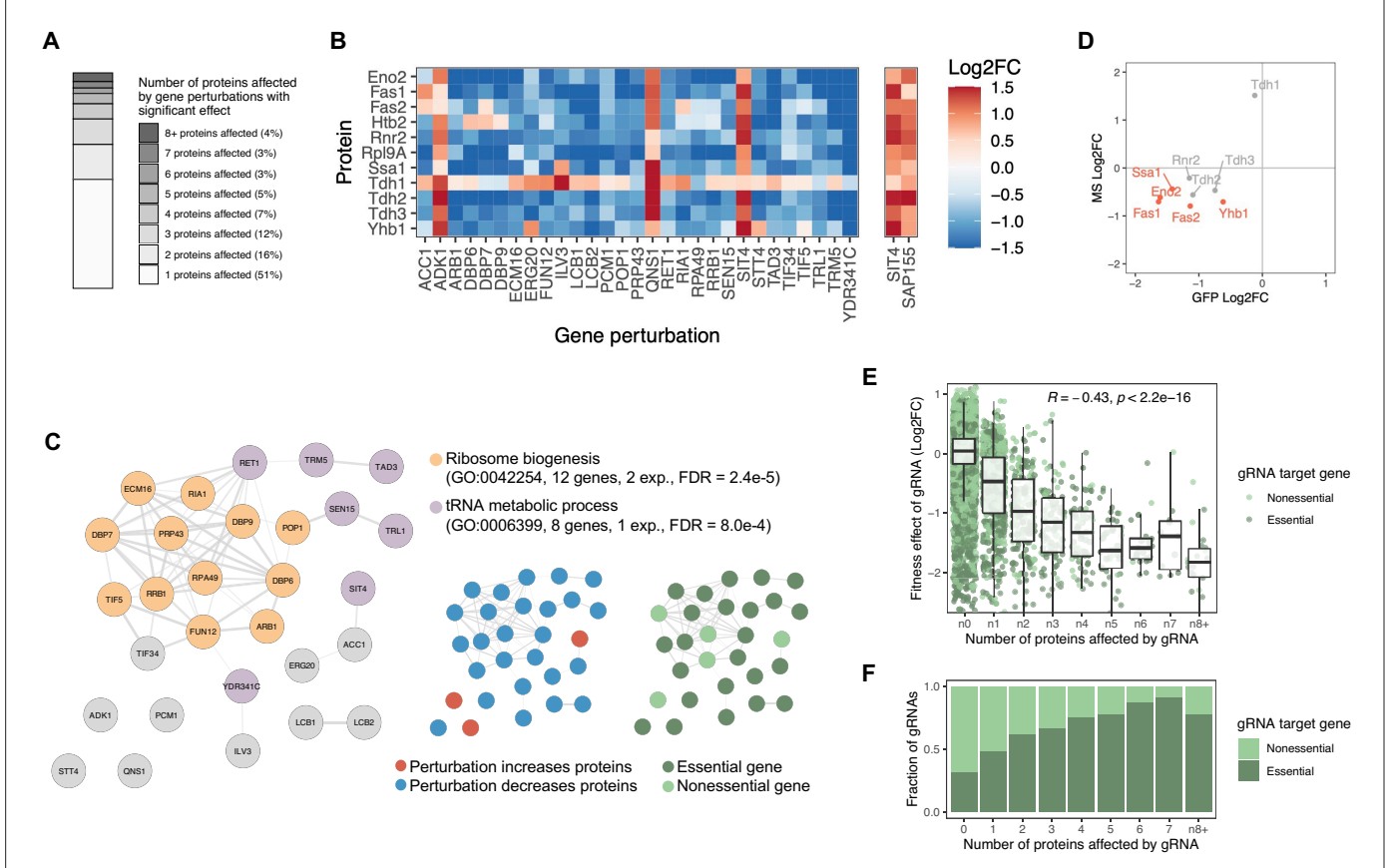

**Figure 5.** Gene perturbations with broad effects on protein abundance. (**A**) Number of proteins significantly affected by each gene perturbation (FDR<0.05). The entire bar represents all gene perturbations with at least one significant effect on a protein. (**B**) Heatmap of protein-level effects of the 29 gene perturbations with broad effects (left panel) and of *SIT4* and *SAP155* gene perturbations for direct comparison (right panel). The color scale is capped at −1.5 and 1.5. (**C**) Network representation of the 29 genes with broad effects. Edges of the network reflect the overall confidence score for protein-protein interactions reported in the STRING database and colors represent enriched functional categories (orange: ribosome biogenesis, purple: tRNA metabolism) (*Doncheva et al., 2018*; *Szklarczyk et al., 2020*). Miniature networks are colored by the prevalent direction of effect on protein abundance (blue: decreased abundance, red: increased abundance) or by gene essentiality (light green: nonessential, dark green: essential). (**D**) Validation of observations from the base editor screen ('GFP') by LC-MS proteomics ('MS') for a hypomorphic mutation in *POP1* (H642Y). Red data points reflect proteins that change significantly in both experiments upon *POP1* perturbation (FDR<0.1). (**E**) For each gRNA, the fitness effect (vertical axis) is plotted against the number of proteins significantly affected by each gRNA (FDR<0.05) (horizontal axis). For the fitness analysis, the 24 hr base editor induction period with galactose was followed by 24 hr of growth in glucose media. Box plot overlays summarize the underlying data points (center line: median; box limits: first and third quartiles; whiskers: 1.5-fold interquartile range). The coloring of the data points reflects whether the gRNA targets an essential or nonessential gene. (**F**) The fraction of gRNAs targeting essential and nonessential genes as a function of the number of proteins significantly affected by each gRNA (FDR<0.05). Note that at the reference time point of the fitness experiment, there was no bias in read counts across the gRNAs affecting different numbers of proteins (data not shown).

The online version of this article includes the following source data and figure supplement(s) for figure 5:

**Source data 1.** Results of fitness screen with 16,452 gRNAs.

**Figure supplement 1.** Proteomics analysis of *POP1* and *SIT4* mutants represented as volcano plot.

**Figure supplement 2.** Base edits and natural variants across the *POP1* gene.

---

All but three of the 29 gene perturbations with broad effects on protein abundance generally lowered protein abundance (*Figure 5B and C*, middle panel). One of the three genes whose perturbation increased protein abundance is the PP2A-like phosphatase *SIT4*. Perturbation of *SAP155*, a gene strictly required for the function of *SIT4*, has very similar effects on protein abundance (*Figure 5B*, right panel) (*Luke et al., 1996*). LC-MS proteomics of a *SIT4* loss-of-function mutant (Q184*) revealed the induction of a large number of proteins involved in cellular respiration, carbohydrate metabolism, and oxidative stress (FDR <0.01, *Figure 5—figure supplement 1B* and *Figure 4—source data 1*).

To better understand how the genetic effects on protein abundance relate to cellular phenotypes, we performed a fitness screen with the same gRNA library used for the protein screen (*Figure 5—source data 1*). We observed that gene perturbations with broader effects on protein abundance were more likely to impair cellular fitness (Pearson's $R^2$=0.18, p<2.2e-16) (*Figure 5E*). Consistent with this finding, these perturbations were also more likely to be in essential genes (*Figure 5F*). Because essential genes are often involved in maintaining general cellular physiology, their perturbation is expected to have more widespread effects on protein abundance than perturbation of nonessential genes that may function in niche processes. This is also in line with earlier observations that essential genes are more interconnected in protein-protein interaction networks (network hubs), and that there is a negative correlation between the connectivity of a gene product and the change in cellular growth rate upon gene deletion (*Batada et al., 2006*; *Jeong et al., 2001*).

Overall, our results suggest that most genetic perturbations with very broad effects on protein abundances are involved in protein biosynthetic functions, as exemplified by *POP1*. However, we also see that perturbation of genes such as *SIT4* and *SAP155* can lead to widespread proteomic changes, in this case via cell physiological adaptations involving carbon metabolism and aerobic respiration. Irrespective of the precise mechanisms, genetic perturbations with broad effects typically involve essential genes and almost always reduce cellular fitness.

Several of the gene perturbations that affect multiple proteins overlap previously identified hotspots of gene expression and protein quantitative trait loci (eQTLs and pQTLs) in *S. cerevisiae* (*Albert et al., 2018*; *Albert et al., 2014*). Thus, the base editor screen identified candidate causal genes for 24 eQTL and pQTL hotspots (*Supplementary file 5*). One such causal gene candidate is *POP1*, which overlaps a protein regulatory hotspot on chromosome 14 (*Albert et al., 2014*) and harbors several coding variants in divergent yeast strains (*Figure 5—figure supplement 2*). This example showcases that targeted genetic screens, especially with methods such as the CRISPR base editor screen that give nucleotide-level resolution, can provide insights into how natural genetic variants affect gene and protein regulation.

## Distinct responses of GAPDH isoenzymes to gene perturbations suggest functional diversification

We next compared the effects of gene perturbations on the three GAPDH isoenzymes, Tdh1, Tdh2, and Tdh3. GAPDH is a highly conserved enzyme best known for its central role in glycolysis and gluconeogenesis, but it has also been implicated in many non-metabolic processes (*Kosova et al., 2017*; *Ringel et al., 2013*; *White and Garcin, 2017*). In *S. cerevisiae*, Tdh3 is thought to be the main GAPDH enzyme in glycolysis and is one of the most highly expressed proteins (*Newman et al., 2006*). None of the three GAPDH genes are essential for cell viability, but a functional copy of either *TDH2* or *TDH*3 is required; in contrast, cells with a *TDH1* deletion, either alone or in combination with a *TDH2* or a *TDH3* deletion, are viable (*McAlister and Holland, 1985b*; *McAlister and Holland, 1985a*; *Randez-Gil et al., 2019*). The three isoenzymes were shown to selectively change abundance in response to available carbon sources and other environmental perturbations (*Casanovas et al., 2015*; *Costenoble et al., 2011*; *Murphy et al., 2015*; *Valadi et al., 2004*). However, despite the key role of GAPDH in metabolism and other cellular processes, the specific functional roles and regulatory mechanisms of the three isoenzymes are still largely unknown.

Our base editor screen revealed that abundance of Tdh1, Tdh2 and Tdh3 responds similarly to the perturbation of many genes, including those involved in central carbon metabolism and biosynthesis of aromatic amino acids (Aro1/2) (*Figure 6A*). However, some gene perturbations have distinct effects on the three isoenzymes. For example, perturbation of genes encoding two members of the Mediator Cdk8 module, Ssn3 and Srb8, affect abundance of Tdh1 and Tdh2 but not Tdh3 (*Figures 6A and 4D*). Ssn3 and Srb8 have been implicated in carbon catabolite repression, a mechanism to repress genes involved in the use of alternate carbon sources when a preferred carbon source such as glucose is available (*Kayikci and Nielsen, 2015*). Because Tdh3 is thought to be the main enzyme responsible for glycolytic flux under high-glucose conditions, it is plausible that the alternative enzymes Tdh1 and Tdh2 are under carbon catabolite repression and play a role when glucose is unavailable. This hypothesis is consistent with strong induction of Tdh1 in the absence of glucose (*Casanovas et al., 2015*; *Murphy et al., 2015*).

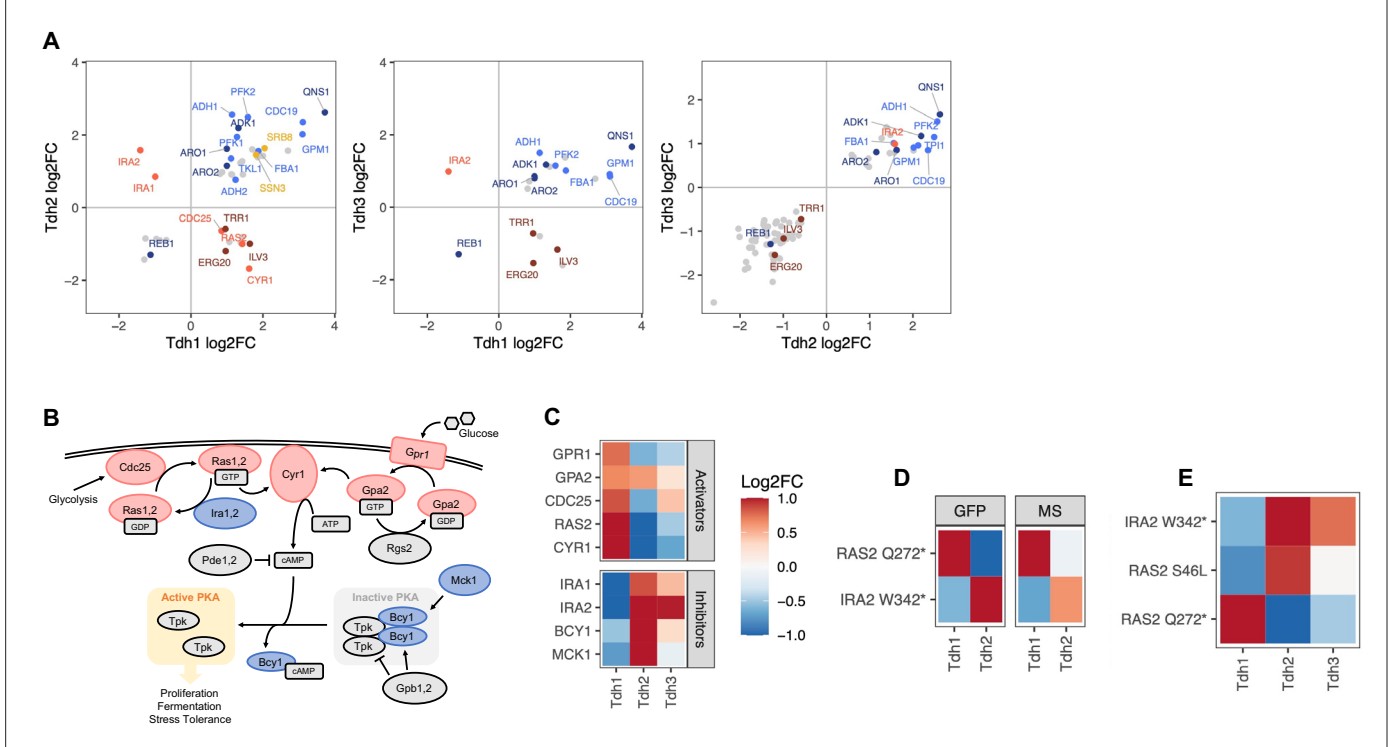

**Figure 6.** Regulators of the three GAPDH isoenzymes Tdh1, Thd2, and Tdh3. (**A**) Pairwise comparison of effects of gene perturbations on Tdh1, Tdh2, and Tdh3 protein abundances. Only genes that upon perturbation affect both Tdh proteins significantly are shown (FDR<0.05). (**B**) Ras/cAMP/PKA pathway representation with components colored if they showed a significant signal for one or more of the GAPDH isoenzymes (red: activators, blue: inhibitors). The scheme in (B) has been adapted from Figure 1 from *Peeters et al., 2017*, with added Mck1 (*Griffioen et al., 2003*; *Peeters et al., 2017*). (**C**) Heatmap representing all genes involved in the PKA pathway that upon perturbation affect at least one of the three GAPDH isoenzymes significantly (FDR<0.1). (**D**) Effects of *IRA2* W342* and *RAS2* Q272* mutations on Tdh1 and Tdh2 protein abundance observed in the base editor screen ('GFP') and by LC-MS-based proteomics on the individual mutants ('MS'). (**E**) Effects of *IRA2* W342*, *RAS2* Q272* and *RAS2* S46L mutations on Tdh1, Tdh2, and Tdh3 protein abundance observed in the base editor screen. The heatmap color scale is identical for C, D, and E, and is capped at −1 and 1.

Tdh1 protein abundance shows the most distinct responses to gene perturbations (*Figure 6A*). The most striking differences occur in response to perturbation of the Ras/cAMP/PKA pathway, a key driver of cell growth and proliferation in response to glucose availability and an integrator of stress responses (*Conrad et al., 2014*; *Dechant and Peter, 2008*; *Smith et al., 1998*; *Figure 6A and B*). Perturbations of activating components of this pathway (Gpr1, Ras2, Cyr1, Cdc25) increase abundance of Tdh1 but decrease abundance of Tdh2 (and, to a lesser extent, Tdh3). Perturbation of the farnesyl pyrophosphate synthetase Erg20 had the same effects. Erg20 is an essential enzyme involved in protein farnesylation. It has been shown that Ras2 farnesylation is critical for PKA activation, suggesting that the signal we see for Erg20 is also the result of a perturbed Ras/cAMP/PKA pathway (*Bhattacharya et al., 1995*). Conversely, perturbations of inhibiting components of this pathway (Ira1/2, Bcy1, Mck1) decrease abundance of Tdh1 while increasing abundance of Tdh2 and Tdh3 (*Figure 6C*). We used LC-MS proteomics to confirm these findings in strains carrying engineered loss-of-function mutations in an inhibitor (*IRA2*) and an activator (*RAS2*) of the pathway (*Figure 6D* and *Figure 4—source data 1*). A functional connection between the Ras/cAMP/PKA pathway and Tdh1/Tdh2 is further supported by the negative genetic interactions between *TDH1/TDH2* and *RAS2* reported in large-scale studies (*Costanzo et al., 2016*; *Costanzo et al., 2010*; *Oughtred et al., 2021*; *Srivas et al., 2016*). The distinct regulatory mechanisms of the three GAPDH isoenzymes uncovered by the base editor screen suggest that they play separate roles in different metabolic states of the cell.

## Discordant gRNA effects point to a gain-of-function mutation in *RAS2*

Among the gRNAs targeting *RAS2*, one is predicted to introduce the missense mutation S46L and another to introduce the loss-of-function mutation Q272*. We observed that these gRNAs had

discordant effects on protein abundance. Ras2 Q272* increased abundance of Tdh1 and decreased abundance of Tdh2, while Ras2 S46L had the opposite effects (*Figure 6E*). The effect of the Ras2 S46L mutation on Tdh1 and Tdh2 was similar to that of a loss-of-function mutation of Ira2, a direct antagonist of Ras2 (*Figure 6E*). Our observation that the S46L mutation activates Ras2 is consistent with the results of saturation mutagenesis of the highly conserved human K-Ras gene, which showed increased activity upon mutation of the homologous serine residue (S39 in human K-Ras) to leucine or valine (*Bandaru et al., 2017*). Human Ras genes are important oncogenes, with activating point mutations found in as many as 20% of all human tumors (*Downward, 2003*). Mutations of this specific serine residue have been found in cervical cancers (COSMIC database, *Tate et al., 2018*). Thus, base editor screens in yeast can recover both loss-of-function and gain-of-function mutations in genes that are relevant to human diseases.

## Discussion

Systems genetics is concerned with how genetic information is integrated, coordinated, and ultimately transmitted through molecular, cellular, and physiological networks to enable the higher order functions of biological systems (*Nadeau and Dudley, 2011*). Proteins are the key molecular players functionally linking genotype and phenotype of a cell. Therefore, systematically studying regulators of protein abundance is crucial to reaching a more complete, system-level understanding of molecular cell physiology. We developed a high-resolution CRISPR/Cas9 base editor screen and used it to investigate how the abundance of a variety of selected yeast proteins is affected by each of tens of thousands of genetic perturbations. Thereby, we gained insights into the extensive regulatory network underlying a cell's proteome, in which the abundance of any given protein is affected by a large number of genetic loci. We found that among the eleven proteins studied here, those involved in core cellular functions respond differently to genetic perturbations than those involved in stress responses. Furthermore, proteins were generally more likely to be affected by perturbations of essential genes than of nonessential genes. Genes affecting the abundance of many of the eleven proteins simultaneously (i.e., hubs of the protein regulatory network) were often involved in post-transcriptional protein biosynthetic processes. In contrast, perturbations that affected only specific proteins were enriched in transcriptional regulation. In addition, we discovered numerous new regulatory relationships, including a new link between the Ras/PKA pathway and the GAPDH isoenzymes that suggests functional diversification of these homologs which enables cells to adapt to different metabolic states.

Earlier large-scale studies on the effects of genetic perturbations on gene expression mostly focused on mRNA levels as a readout. For example, a study by Kemmeren and colleagues examined the effects of 1484 gene deletions on the *S. cerevisiae* transcriptome (*Kemmeren et al., 2014*). The effects of gene perturbations on mRNA levels in that study and on protein levels in our study agree well when only effects that reach statistical significance in both studies are considered (*Figure 3—figure supplement 2*). Gene perturbations with detected effects on mRNA but not protein levels are enriched in genes with a role in "chromatin organization" (FDR=0.01), suggesting that perturbations of such genes tend to affect mRNA levels, but are then buffered and do not lead to altered protein levels. It is difficult to draw further conclusions from the comparison between these two studies because the genes perturbed in the Kemmeren et al. study were selected to play a role in gene regulation. Furthermore, while there is a low false positive rate in our study, the false negative rate is likely to be high due to limited statistical power and because gRNA target residues are not always edited efficiently, and edits may not have strong effects on gene function. Statistical power in protein abundance screens can be boosted by increasing the number of cells sorted for each GFP bin, as well as by increasing sequencing depth. Studies that assess effects of gene perturbations on both mRNA and protein levels simultaneously are also becoming increasingly feasible thanks to recent developments in single-cell CRISPR-based screening and decreasing sequencing costs (*Przybyla and Gilbert, 2022*).

This and previous studies show that CRISPR base editing outcomes are often heterogeneous and incomplete (*Figure 1B*). In genetic screens that use the CRISPR base editor, the heterogeneity of the mutant population generated by each gRNA should be kept in mind when interpreting the data. We would argue, however, that in many cases, base editing is sufficiently specific and localized to obtain insights into the effects of mutations in the targeted genomic area. For example, in our test screens for canavanine resistance and fitness, in which we used gRNAs predicted to introduce stop codons into the CAN1 gene and into essential genes, respectively, we observed the expected loss-of-function

effects for a majority of gRNAs tested (*Figure 2*). In the canavanine screen, we also observed that gRNAs predicted to introduce missense mutations at highly conserved residues were more likely to lead to a loss of function than gRNAs predicted to introduce missense mutations at less conserved residues, further highlighting that informative results that can be obtained with the base editor despite the heterogeneity in editing outcomes. However, it is imperative to confirm key findings with independently generated mutations, as we did in this study.

Another consideration with CRISPR base editor screens is the potential for off-target editing. Completely random off-target activity of the base editor is of little concern because it is not expected to generate statistically significant signals. Reproducible off-target editing by a gRNA at a site other than the intended target site would, however, be problematic. Such off-target editing can be minimized by excluding gRNAs that have similar target sequences elsewhere in the genome. In this study, we excluded gRNAs for which more than one target site in the genome matched the twelve nucleotides in the seed region (directly upstream of the PAM site) (*DiCarlo et al., 2013*). We observe that most off-target editing occurs just outside the target window, and it is seen at much lower frequency than on-target editing in the target window (*Figure 1*, *Figure 1—figure supplement 2*). The possibility of off-target effects is another reason to validate key findings with independently generated mutations. New base editors with improved accuracy and efficiency have the potential to improve future genetic screens (*Anzalone et al., 2020*).

The CRISPR base editor, by binding its target sequences within or upstream of genes or transcripts, could interfere with transcription and translation and thereby elicit effects that are independent of the desired base edits (*Liu et al., 2016b*, *Qi et al., 2013*). To reduce confounding signals due to such unintended interference, we placed the gene encoding the base editor under a galactose-inducible promoter (*DiCarlo et al., 2013*). After base editing in the presence of galactose, we grew the edited cultures for multiple generations in the absence of galactose. By the time the cultures are sampled for phenotyping, base editor levels are expected to be sufficiently reduced to no longer interfere with gene expression. This expectation is supported by our ability to detect opposing effects of gRNAs introducing different types of mutations into the same gene (e.g., the two mutations leading to loss-of-function and gain-of-function of RAS2); such observations would not be possible in the presence of strong gene-wide transcriptional or translational interference effects.

Future studies will be able to build on the experimental framework described here to further refine and complete our understanding of the genetic factors influencing protein abundance. Potential extensions include screening for abundance of additional proteins, examining the effects of genetic perturbations on other molecular (e.g., mRNA abundances, as discussed above) and physiological phenotypes, increasing the number of gRNAs and using alternative base editors to achieve denser sampling of mutations across coding and non-coding regions, introducing multiple gRNAs per cell to investigate interactions between genetic perturbations, and performing screens in different genetic backgrounds or under different growth or stress conditions. A dense genome-wide sampling of mutations, together with quantitative molecular and cellular phenotypic readouts across different environmental conditions and genetic backgrounds, will provide the data needed to build holistic models that bring us closer to predicting the functional consequences of altering any base pair in the genome through natural sequence variation or engineered perturbation (*Kinney and McCandlish, 2019*).

## Methods
### Material availability
Important plasmids generated in this study are deposited in Addgene (Base editor plasmid (pGal-BE3): #172409; gRNA plasmid with 2 kb placeholder (pgRNA-backbone): #172517).

### Resource availability
The DNA sequencing read data is deposited in the NCBI Sequence Read Archive under BioProject ID PRJNA732764 and the mass spectrometry proteomics data is deposited in the ProteomeXchange Consortium partner repository PRIDE with the dataset identifier PXD029363 (*Perez-Riverol et al., 2018*). All custom code is available on GitHub without restrictions: https://github.com/OlgaTSchubert/ProteinGenetics (copy archived at swh:1:rev:bab9abdb34f8ab883e5881f9d9a24abfdd1e0a6c; *Schubert, 2022*).

## Strains and cultures

*Saccharomyces cerevisiae* S288C strain BY4741 (MATa his3Δ1 leu2Δ0 met15Δ0 ura3Δ0) was used for all experiments except the protein abundance screen (*Brachmann et al., 1998*). For that, we used eleven strains from the Yeast GFP Library, which is also based on BY4147. In each of these strains, a GFP cassette (GFP(S65T)-HIS3MX6, where HIS3 is from *Lachancea kluyveri*) is fused in-frame to the C-terminus of a gene of interest (*Huh et al., 2003*). As a reference genome, we used the *S. cerevisiae* S288C genome sequence version R64-2-1 from the *Saccharomyces* Genome Database (SGD).

All yeast strains generated in this study are listed in *Supplementary file 2*.

Yeast cultures were grown at 30 °C with 250 rpm orbital shaking for liquid cultures. Either of the following media was used:

- Rich media (**YPD**):
  - 20 g/L Bacto Peptone (BD Biosciences #211820)
  - 10 g/L Bacto Yeast Extract (BD Biosciences #212720)
  - 2 g/L dextrose/glucose (MP Biomedicals #901521)
- Synthetic (CSM) dropout media lacking leucine and uracil (**CSM -Leu -Ura**):
  - 6.7 g/L Difco's YNB without Amino Acids (BD Biosciences #291940)
  - 2 g/L dextrose/glucose (MP Biomedicals #901521)
  - 0.67 g/L CSM powder lacking leucine and uracil (Sunrise Science Products #1038–100)
  - For non-selective version of CSM, 100 mg/L L-leucine (Sigma #L8912) and 20 mg/L uracil (Sigma #U0750) were added.
- Synthetic (SC) dropout media lacking leucine and uracil (**SC -Leu -Ura**):
  - 6.7 g/L Difco's YNB without Amino Acids (BD Biosciences #291940)
  - 2 g/L dextrose/glucose (MP Biomedicals #901521)
  - 1.74 g/L SC dropout powder lacking leucine and uracil (Sunrise Science Products #1318–030)
  - For non-selective version of SC, 100 mg/L L-leucine (Sigma #L8912) and 20 mg/L uracil (Sigma #U0750) were added.
- Canavanine media:
  - 6.7 g/L Difco's YNB without Amino Acids (BD Biosciences #291940)
  - 2 g/L dextrose/glucose (MP Biomedicals #901521)
  - 20 mg/L L-histidine hydrochloride monohydrate (Fisher Scientific #BP383100)
  - 20 mg/L L-methionine (Sigma #M9625)
  - 20 mg/L adenine sulfate salt (Sigma #A2545)
  - 60 mg/L canavanine sulfate salt (Sigma #C9758)
  - Note that canavanine media must not contain arginine for the canavanine resistance selection to work.

Agar plates were prepared with the same media formulation but with agar added at 20 g/L (BD Biosciences #214030). For induction of the base editor, which is under a galactose-inducible promoter, we used CSM/SC media with 2 g/L galactose (Sigma #G0625) instead of dextrose/glucose. Cryostocks were prepared by mixing yeast cultures or cell suspensions with sterile 40% glycerol-in-water solution to a final concentration of 13.3% glycerol (v/v) and were stored at –80 °C.

For the construction of individual plasmids and small libraries, we used a DH5α-derived *Escherichia coli* strain (NEB #C2987). For large library transformations, we used E. cloni 10 G Supreme Electrocompetent Cells (Lucigen #60081). Liquid cultures were grown at 37 °C with 250 rpm orbital shaking. If not mentioned otherwise, both solid and liquid cultures were grown in LB media (20 g/L, Fisher Scientific #BP1427), which for selections was supplemented with 100 µg/ml ampicillin (Sigma-Aldrich #A0166) and/or 50 µg/ml kanamycin (Fisher Scientific #BP906). Cryostocks were prepared by mixing bacterial cultures or cell suspensions with sterile 40% glycerol-in-water solution to a final concentration of 20% glycerol (v/v) and were stored at –80 °C.

## Plasmids

All plasmids generated in this study are listed in *Supplementary file 3* and all primers used are listed in *Supplementary file 4*.

**Base editor plasmid (pGal-BE3)** (*Figure 1—figure supplement 1A*). Our yeast base editor plasmid was constructed based on the yeast Cas9 plasmid described by DiCarlo and colleagues (Addgene #43802) (*DiCarlo et al., 2013*). In addition to the Cas9 gene under a galactose-inducible *GALL* promoter (*Mumberg et al., 1994*), a CEN/ARS origin, and other elements, this plasmid encodes *LEU2* which enables its selection in leucine-auxotroph yeast grown in media lacking leucine (*Mumberg*

*et al., 1995*). Using Gibson Assembly, we swapped the Cas9 gene encoded on the plasmid with the base editor v3 (BE3) described by Komor and colleagues (Addgene #73021) (*Komor et al., 2016*). This plasmid is deposited in Addgene: #172409.

**gRNA plasmid with 2 kb placeholder (pgRNA-backbone)** (*Figure 1—figure supplement 1B*). Our yeast gRNA plasmid was constructed based on the yeast gRNA plasmid described by DiCarlo and colleagues (Addgene #43803) (*DiCarlo et al., 2013*). In addition to the gRNA coding sequence under an SNR52 promoter (for RNA Polymerase III transcription), a 2μ origin, and other elements, this plasmid encodes *URA3* which enables its selection in uracil-auxotroph yeast grown in media lacking uracil (*Mumberg et al., 1995*). Using Gibson Assembly, we swapped the gRNA targeting sequence encoded on the plasmid with the guide placeholder cassette from the lentiCRISPR v2 plasmid described by Sanjana and colleagues (Addgene #52961) (*Sanjana et al., 2014*). The 2 kb guide place-holder present in this cassette and flanked by Esp3I/BsmBI restriction sites facilitates the construction of individual gRNA plasmids as well as gRNA plasmid libraries. All other Esp3I/BsmBI sites present in the plasmid backbone were removed by site directed mutagenesis (Agilent #210515). This plasmid is deposited in Addgene: #172517.

**gRNA plasmids for characterization and validations**. Individual gRNA plasmids were generated based on the pgRNA-backbone plasmid described above according to the protocol by Joung and colleagues (*Joung et al., 2017*). Briefly, two partially complementary 25-nucleotide oligonucleotides were annealed, forming the gRNA targeting sequence. The resulting double-stranded DNA fragment with 4-nucleotide overhangs was then inserted into the pgRNA-backbone plasmid by Golden Gate Assembly (*Engler et al., 2008*) using the Esp3I/BsmBI restriction enzyme (NEB #E1602S). All gRNA targeting sequences for these individual base editing experiments are listed below.

**gRNA plasmid libraries**. The barcoded gRNA plasmid libraries were generated based on the pgRNA-backbone plasmid described above. Details are described in a separate section below. Briefly, a kanamycin resistance cassette was barcoded with 20 random nucleotides and inserted into the pgRNA-backbone plasmid, resulting in a barcoded plasmid library (*Figure 1—figure supplement 1C*). Subsequently, the oligonucleotide libraries with all gRNA targeting sequences were inserted into this barcoded plasmid library, resulting in barcoded gRNA plasmid libraries.

## Base editor characterization with individual gRNAs

To characterize the base editor efficiency and editing window, we selected seven gRNAs predicted to introduce premature stop codons into *CAN1*, *ADE1* and *ADE2* using Benchling (https://bench-ling.com). The gRNAs were cloned into the pgRNA-backbone plasmid as described above in the 'Plasmids' section. In addition, we tested a gRNA-containing plasmid from DiCarlo and colleagues predicted to introduce a missense mutation into *CAN1* (*DiCarlo et al., 2013*). We first transformed our BY4741 yeast strain with the base editor plasmid pGal-BE3 using a high-efficiency lithium acetate-based protocol described by Gietz and Schiestl with minor modifications (*Gietz and Schiestl, 2007a*). The resulting strain was transformed with the eight gRNA plasmids individually (one gRNA plasmid per strain) according to the same protocol. Successful transformants growing on selective plates (CSM -Leu -Ura) were then transferred into selective galactose media to induce base editor expression and grown at 30°C until they reached stationary phase (44 hr).

Approximately 1e8 cells per culture were harvested and genomic DNA was extracted using Qiagen's DNeasy Blood & Tissue Kit (Qiagen #69506) according to their supplementary protocol "Purification of total DNA from yeast" (DY13 Aug-06) but using 10 U/ml zymolase (amsbio #120491-1) instead of lyticase. The predicted genomic target region of each gRNA in the respective DNA sample was then amplified using PfuUltra II Fusion HS DNA Polymerase (Agilent #600670) with the following thermocycling protocol: 2 min at 95 °C, 22 x (30 s at 95 °C, 30 s at 55 °C, 30 s at 72 °C), 3 min at 72 °C. Primers were designed to include overhangs that mimic the product of the Illumina Nextera transposase in order to facilitate further processing by Illumina sequencing reagents (xxx indicates the 20-nucleotide genomic binding regions of these primers):

- Forward Nextera primers: TCGTCGGCAGCGTCAGATGTGTATAAGAGACAGxxx
- Reverse Nextera primers: GTCTCGTGGGCTCGGAGATGTGTATAAGAGACAGxxx

The resulting fragments were pooled, then purified with the MinElute PCR Purification Kit (Qiagen #28004). A second round of PCR to extend the Nextera adapters was performed using reagents from the Nextera DNA Library Prep Kit (NPM, PPC and indexes; Illumina #FC-121-1031) according

to the manufacturer's instructions. This second PCR product was run on a 2% agarose gel and gel-extracted using a QIAquick Gel Extraction Kit (Qiagen #28704). The amplicon library was quantified by qPCR using the KAPA Library Quantification Kit (KAPA/Roche #KK4824). To increase diversity, 5% PhiX Sequencing Control v3 (Illumina #FC-110-3001) was added. The final library was brought to a concentration of 15 pM and sequenced on an Illumina MiSeq with paired-end 150-base pair reads.

The sequencing data was processed using CRISPResso2 (v2.0.45) (*Clement et al., 2019*) using the default settings, except for the following parameters: --min_average_read_quality 30, --quantification_window_size 20, --quantification_window_center -10, --base_editor_output. To determine the fraction of all possible base edits, the CRISPResso2 output file "Quantification_window_nucleotide_frequency_table.txt" was processed with a custom R script (R version 4.0.3, RStudio version 1.3).

## Base editing rate

To assess how long after induction of the base editor the majority of cells would contain the desired edit, we designed two gRNAs targeting loss-of-function mutations into GFP (Q183*/Q184* and E222K) (*Fu et al., 2015*). As a control, we used a gRNA that targets the *CAN1* gene (W174*), which is expected to not have an effect on GFP fluorescence. The gRNAs were cloned into the pgRNA-backbone plasmid and transformed into a BY4741-derived yeast strain previously transformed with the base editor plasmid pGal-BE3 as described above. The strain came from the GFP collection and contained a genomically inserted GFP translationally fused to *HXK2* (*Huh et al., 2003*). Successful transformants were grown overnight in selective media (CSM -Leu -Ura) and then transferred into selective galactose media to induce base editor expression. At 0 hr, 12 hr, and 24 hr, cells were centrifuged, resuspended in cold PBS, and kept on ice until analysis. Once all samples were collected, GFP fluorescence of the cells was measured by flow cytometry.

## gRNA library design

Using custom R scripts run in RStudio (R version 4.0.3, RStudio version 1.3), we first compiled a list of all possible gRNA targeting sequences (guides) suitable for base editing in yeast by searching for all possible guides in the yeast genome, defined as the 20 nucleotides upstream of an NGG PAM site. We also added 10,000 guides not targeting the yeast genome for use as controls. We predicted the expected base editing outcome assuming that all Cs will be converted to Ts within a window 13–17 base pairs upstream of the PAM site (*Komor et al., 2016*). We then excluded (i) guides with a potential RNA Polymerase III terminator (TTTT), (ii) guides that target regions where two or more annotated genes overlap, and (iii) guides whose target region harbors homology to another genomic location and could therefore cause unexpected off-target effects (considering all perfect matches in the 12-nucleotide seed region directly upstream of the PAM) (*DiCarlo et al., 2013*; *Jinek et al., 2012*).

For all missense mutations expected to be introduced by the gRNAs, we determined the PROVEAN score, which predicts the effect of amino acid substitutions based on conservation (*Choi et al., 2012*). For gRNAs expected to introduce multiple missense mutations, we chose as representative value the maximal absolute PROVEAN score. We then defined an absolute PROVEAN score of >5 as highly conserved (18% of all gRNAs predicted to introduce missense mutations into the yeast genome and passing the above mentioned filters).

From the filtered gRNA database we extracted five different sets of gRNAs:

- CA: All 170 gRNAs predicted to introduce missense or nonsense mutations in *CAN1* and *ADE2*; only the 91 *CAN1* targeting gRNAs were used for experiments reported here.
- ES: All 5442 gRNAs predicted to introduce premature stop codons in essential genes.
- NS: 5500 gRNAs predicted to introduce premature stop codons in nonessential genes, randomly selected out of 19,772 gRNAs with that property.
- EP: 5500 gRNAs predicted to introduce missense mutations at highly conserved residues in essential genes, randomly selected out of 8955 gRNAs with that property.
- NP: 5500 gRNAs predicted to introduce missense mutations at highly conserved residues in nonessential genes, randomly selected out of 33,357 gRNAs with that property.

We also designed three sets of control gRNAs predicted to not introduce any (coding) mutations in the yeast genome: 500 gRNAs introducing synonymous mutations in yeast genes, 500 gRNAs without a C in the targeting region, and 500 gRNAs with random sequences not targeting the yeast genome. All gRNAs are listed in *Supplementary file 1*.

For the synthesis of the gRNA libraries, each gRNA targeting sequence of 20 nucleotides was flanked by 30-nucleotide homology arms on either side for Gibson Assembly into the receiver plasmid (pgRNA-backbone) (*Figure 1—figure supplement 1*). Additionally, at both ends of the oligonucleotides 15-nucleotide primer binding sites were added which differed for each of the five sets of gRNAs described above (control gRNAs were given the same primer binding sites as the ES gRNAs). These primer binding sites allow for specific amplification of individual sublibraries from the main library. The total length of the resulting oligonucleotides was 122 nucleotides. Note that we originally planned to remove the primer binding sites with an MluI restriction digest and therefore excluded from synthesis a small number of gRNAs that contained MluI sites in the targeting sequence. However, we ended up using an alternative strategy not dependent on MluI (see below).

All 23,541 gRNAs were synthesized as a single oligonucleotide library (Agilent #G4893A).

## Construction of barcoded gRNA plasmid libraries

The barcoded gRNA plasmid library was cloned based on the pgRNA-backbone plasmid described above. First, a kanamycin resistance cassette (originally likely from the pCR-Blunt II-TOPO plasmid, Thermo Fisher Scientific #451245) was amplified using a primer pair of which the forward primer (OS123) contained an overhang with 20 random nucleotides. After removal of the plasmid template by DpnI digest (NEB #R0176S) and DNA purification (Zymo Research #D4013), this barcoded resistance cassette was introduced into the PCR-amplified, DpnI-treated, and purified pgRNA-backbone plasmid backbone by Gibson Assembly (NEB #E5510S) (*Gibson et al., 2009*). After DNA purification (Zymo Research #D4013), plasmid libraries were eluted in 10 µl ultra pure water and 2 µl was electroporated into five high-efficiency electrocompetent *Escherichia coli* cell aliquots each (Lucigen #60081) using a MicroPulser Electroporator (Bio-Rad Laboratories #1652100). After 60 min of growth in liquid recovery media (SOC outgrowth media, NEB #B9020S), cells were plated onto five 150 mm LB agar plates supplemented with ampicillin/carbenicillin (100 µg/ml) and kanamycin (50 µg/ml) and incubated overnight at 37 °C. Serial dilution platings and colony counts performed in parallel indicated a total of 150,000 independent transformants, each assumed to contain a plasmid with a different barcode. All colonies were then washed off the plates, pooled and subjected to plasmid extraction (Qiagen #12963).

The oligonucleotide library consisting of the five different sublibraries described above under 'gRNA library design' arrived lyophilized and was first dissolved in Tris-EDTA buffer (10 mM Tris-HCl, 0.1 mM EDTA, pH 8) to a concentration of 100 nM. An aliquot was then further diluted to 10 nM and the desired sublibraries were amplified by a 15-cycle PCR with sublibrary-specific primers (Q5 High-Fidelity 2 X Master Mix by NEB #M0492S). After DNA purification (Zymo Research #D4013), a second 5-cycle PCR was performed with primers binding more proximal to the gRNA targeting sequences than the previous primers, thereby excluding the library-specific primer binding sites from the final oligonucleotide libraries. This second PCR was done with the same polymerase and again followed by DNA purification.

The amplified oligonucleotide sublibraries were then cloned into the barcoded pgRNA-backbone plasmid library described above according to the protocol by Joung and colleagues (*Joung et al., 2017*). First, the barcoded plasmid library was digested with Esp3I for 60 min at 37 °C (Thermo Fisher Scientific #FD0454) and gel-purified (Zymo Research #D4007). The resulting linearized plasmid backbone was then combined with one of the amplified oligonucleotide libraries and assembled into a final gRNA plasmid library by Gibson Assembly (NEB #E5510S) (*Gibson et al., 2009*). After DNA purification (Zymo Research #D4013), plasmid gRNA libraries were eluted in 8 µl ultra pure water and 2 µl was electroporated into high-efficiency electrocompetent *Escherichia coli* (Lucigen #60081) using a MicroPulser Electroporator (Bio-Rad Laboratories #1652100). After 60 min of growth in liquid recovery media (Lucigen), *E. coli* cells were plated onto a 150 mm LB agar plate supplemented with ampicillin/carbenicillin (100 µg/ml) and kanamycin (50 µg/ml) and incubated overnight at 37 °C. Serial dilution platings and colony counts performed in parallel indicated a total of 1–2 million independent transformants for each sublibrary. Colonies were then washed off the plates and subjected to plasmid extraction (Qiagen #12963).

## Canavanine screen

For the canavanine screen, we started with the BY4741 strain previously transformed with pGal-BE3 as described in the section 'Base editor characterization with individual gRNAs'. We transformed

this strain with a gRNA plasmid library based on the CA gRNA set using a large-scale high-efficiency lithium acetate-based protocol described by Gietz and Schiestl with minor modifications (*Gietz and Schiestl, 2007b*). We scaled up the transformation 10-fold compared to a single transformation reaction and used 10 µg of plasmid DNA for cells from a 50 ml culture at an $OD_{600}$ of 0.7 (3.5e7 cells/ml). The heat shock at 42 °C lasted 40 min. After the transformation, we plated the cells onto nine selective 100 mm plates (CSM -Leu -Ura). Across all plates, there were approximately 150,000 colonies, which were washed off, pooled and frozen in 20% glycerol at –80 °C.

About 5e8 transformed cells were thawed and grown overnight in 100 ml selective glucose media (CSM -Leu -Ura). The next day, 2.5e8 cells were sampled and 5e7 cells were transferred into 100 ml selective galactose media (CSM/gal -Leu -Ura). After 24 hr of growth in galactose, again 2.5e8 cells were sampled ('0 hr') and 1e8 cells were transferred into 200 ml canavanine media (60 mg/L L-canavanine sulfate). From the canavanine-treated culture, 2.5e8 cells were sampled after 24 and 48 hr.

Plasmid DNA was extracted from the sampled cells using the Zymoprep Yeast Plasmid MiniPrep II kit (Zymo Research #D2004) but with a modified first step where cell pellets were spheroblasted with 10 U/ml zymolase (amsbio #120491–1) in sorbitol buffer (1 M Sorbitol, 100 mM EDTA pH 8, 14 mM beta-mercaptoethanol). The plasmid region containing the gRNA target sequence and the barcode was amplified using the KAPA HiFi HotStart ReadyMix (KAPA/Roche KK2602) with the following thermocycling protocol: 45 s at 98 °C, 18 x (15 s at 98 °C, 30 s at 55 °C, 30 s at 72 °C), 1 min at 72 °C. The forward (OS130-139) and reverse (OS125, PAGE-purified) primers included overhangs that mimic the product of the Illumina Nextera transposase as described above. The forward primer also included a stagger sequence of 9–18 nucleotides to introduce diversity in the sequencing library (*Joung et al., 2017*). The resulting fragments were purified (Zymo Research #D4013) and a second round of PCR to extend the Nextera adapters was performed using reagents from the Nextera DNA Library Prep Kit (NPM, PPC and indexes; Illumina #FC-121–1031) according to the manufacturer's instructions. Concentrations of PCR products were determined (Qubit dsDNA HS Assay Kit, Thermo Fisher Scientific #Q32851) and equal amounts of DNA from each sample were combined. The pooled sample was run on a 2% agarose gel and gel-extracted (Zymo research #D4007). The amplicon library was quantified by qPCR using the KAPA Library Quantification Kit (KAPA/Roche #KK4824). To increase diversity, 10% PhiX Sequencing Control v3 (Illumina #FC-110–3001) was added. The final library was brought to a concentration of 15 pM and sequenced on an Illumina MiSeq with paired-end 171- and 131-base pair reads. Note that some guide sequences are cut-off (18 instead of 20 nucleotides) because sequencing reads were not long enough.

## Fitness screen

Note that the fitness screen using the ES gRNA set (*Figure 2C*) was part of a larger experiment using the ES, NS, EP and NP gRNA sets, which was done in duplicate. The fitness screen described later (*Figure 5E and F*) included these two replicates as well as two additional replicates of the same screen performed in parallel using just the NS, EP, and NP gRNA sets (without the ES set).

The library transformations for the fitness screen were also started with the BY4741 yeast strain previously transformed with pGal-BE3 as described in the section 'Base editor characterization with individual gRNAs'. We transformed this strain once with a combined gRNA plasmid library containing the ES, NS, EP, and NP gRNA sets and once with a combined library containing only the NS, EP, and NP gRNA sets, following a large-scale high-efficiency lithium acetate-based protocol with minor modifications (*Gietz and Schiestl, 2007b*). We scaled up each transformation 10-fold compared to a single transformation reaction and used 10 µg of plasmid DNA for cells from a 100 ml culture at an $OD_{600}$ of 0.6 (3e7 cells/ml). The heat shock at 42 °C lasted 90 min. After the transformation, cells were transferred into 100 ml selective media (SC -Leu -Ura) and incubated on a shaker at 30 °C. After 24 hr, cultures were diluted 1:50 and grown until they reached stationary phase. Serial dilution platings and colony counts performed in parallel indicated a total of 2–3 million independent transformants for each of the two transformations.

The fitness screen time-course was done in duplicates for each of the two library-transformed yeast strains, resulting in four quasi-replicates. Four pre-cultures inoculated with 1.3e8 cells were grown overnight in 50 ml selective glucose media (SC -Leu -Ura). The next day, 1.3e8 cells were transferred into 50 ml selective galactose media (SC/gal -Leu -Ura). After 24 hr in galactose, 1.3e8 cells were transferred into 50 ml pre-warmed glucose media (SC -Ura). From then on, every 12 hr 1.3e8 cells

were transferred into 50 ml pre-warmed glucose media. Samples of 1.3e8 cells were harvested at the time of transfer into galactose media as well as 0, 8, and 24 hr after transfer to glucose media. For harvesting, cells were spun down, resuspended in 200 µl PBS (Gibco/Thermo Fisher Scientific #20012027) and stored at –20 °C.

To facilitate plasmid extraction, harvested cells (in 200 µl PBS) were first spheroblasted by adding zymolase (amsbio #120491–1), DL-dithiothreitol (DTT, Sigma-Aldrich #D5545) and RNase A (PureLink, Thermo Fisher Scientific #12091021) to the following final concentrations: 180 U/ml zymolase, 20 mM DTT and 0.2 mg/ml RNase A. The cells were incubated in this solution for 4 hr at 35 °C, followed by 10 min at 99 °C to inactivate zymolase. After cool-down to room temperature, Qiagen's QIAprep Spin Miniprep Kit was used according to the manufacturer's instructions but omitting the first step (Qiagen #27104).

The plasmid region containing the gRNA target sequence and the barcode was amplified using the KAPA HiFi HotStart ReadyMix (KAPA/Roche KK2602) supplemented with EvaGreen (Biotium #31000) using the following thermocycling protocol: 45 s at 98 °C, 16 x (15 s at 98 °C, 30 s at 55 °C, 30 s at 72 °C), 1 min at 72 °C. This was performed on an Agilent AriaMx Real-Time PCR System to be able to monitor and stop the amplification before it started to plateau, which was at 16 cycles. The forward (OS140-149) and reverse (OS125, PAGE-purified) primers included overhangs that mimic the product of the Illumina Nextera transposase as described above. The forward primer also included a stagger sequence of 9–18 nucleotides to introduce diversity in the sequencing library (*Joung et al., 2017*). A second round of PCR (6 cycles) to extend the Nextera adapters was performed using the KAPA HiFi HotStart ReadyMix (KAPA/Roche KK2602) again, together with unique dual indexes (IDT for Illumina DNA/RNA UD Indexes, Integrated DNA Technologies). Concentrations of PCR products were determined by Qubit (dsDNA HS Assay Kit, Thermo Fisher Scientific Q32851) and equal amounts of DNA from each sample were combined. The pooled sample was run on a 2% agarose gel and gel-extracted (Zymo research D4007). The amplicon library was quantified by Qubit (dsDNA HS Assay Kit, Thermo Fisher Scientific Q32851). To increase diversity, PhiX Sequencing Control v3 (Illumina #FC-110–3001) was added and the final library was sequenced on an Illumina NextSeq 500 in HighOutput mode with paired-end 80- and 70-base pair reads.

## Protein abundance screen

For the protein abundance screens, we selected eleven proteins that are involved in various cellular functions but also highly abundant to ensure detectability by flow cytometry. For each selected protein of interest, we streaked out the corresponding GFP-tagged strains from the GFP collection (*Huh et al., 2003*) on YPD plates and from there continued with a single colony from each GFP strain. We first transformed each GFP strain with the base editor plasmid pGal-BE3 using a high-efficiency lithium acetate-based protocol with minor modifications (*Gietz and Schiestl, 2007a*). We then transformed the resulting strains with a combined gRNA library containing the NS, EP and NP gRNA sets, following a large-scale high-efficiency lithium acetate-based protocol with minor modifications (*Gietz and Schiestl, 2007a*). We scaled up each transformation 10-fold compared to a single transformation reaction and used 10 µg of plasmid DNA for cells from a 100 ml culture at an $OD_{600}$ of 0.4 (2e7 cells/ml). The heat shock at 42 °C lasted 90 min. After the transformation, cells were transferred into 100 ml selective media (SC -Leu -Ura) and incubated on a shaker at 30 °C for 24 hr. Subsequently, cultures were diluted ~1:25 and grown until they reached stationary phase. We prepared multiple cryostocks of 2.5e8 cells for each library-transformed GFP strain and stored them at –80 °C until use. Serial dilution platings and colony counts performed in parallel indicated a total of 2–4 million independent transformants for each of the GFP strains.

The protein screen was performed as follows for each GFP strain separately (on consecutive days). First, a cryostock from a library-transformed GFP strain was inoculated in 100 ml selective glucose media (SC -Leu -Ura) and grown overnight. The next day, 7.5e7 cells were centrifuged (1 min at 5000 g), resuspended in 1 ml PBS (Gibco/Thermo Fisher Scientific #20012027) and transferred into 30 ml selective galactose media (SC/gal -Leu -Ura), starting at 2.5e6 cells/ml ($OD_{600}$ of 0.05). After 24 hr, eight replicate cultures were set up in 30 ml pre-warmed glucose media (SC -Ura), each starting at 2.5e6 cells/ml ($OD_{600}$=0.05) from the galactose culture. After 8–9 hr, when the cultures reached mid-exponential growth (~2e7 cells/ml or $OD_{600}$ of 0.4–0.5), they were processed for fluorescence-activated cell sorting (FACS). Briefly, 3 ml of each replicate culture was centrifuged (1 min at 3000 g)

and resuspended in 3 ml cold PBS (Gibco/Thermo Fisher Scientific #20012027), transferred through a 35 µm cell strainer into a 5 ml polystyrene round-bottom test tube (Falcon #352235) and kept on ice from then on. The sorting was performed on a BioRad S3e cell sorter with a cooled sample block. For the sorting, only the 40–50% most similar cells in terms of size and granularity were used (oval gate at densest area on forward scatter (FSC) vs. side scatter (SSC) plot, linear axes). From that FSC-SSC gate, we first sorted 500,000 cells as a control population. Second, we sorted 200,000 cells from the 5% of cells with the highest and the 5% of cells with the lowest GFP signal each (rectangular gates on GFP histogram plot, log axis). Sort speed was set to 10,000 events per second and sorted cells were collected in 5 ml polystyrene round-bottom test tubes (Falcon #352058). The sorted cells (in sheath fluid (PBS), BioRad #12012932) were then transferred into 1.5 ml tubes and centrifuged at full speed for 30 s. Supernatant was carefully removed, leaving ~200 µl behind to minimize disturbing the invisible cell pellet. The sorted and concentrated cell samples were then stored at –20 °C until all sorts for all GFP strains were completed.

To facilitate plasmid extraction, harvested cells (in 200 µl PBS) were first spheroblasted by adding zymolase (amsbio #120491–1), DL-dithiothreitol (DTT, Sigma-Aldrich #D5545) and RNase A (PureLink, Thermo Fisher Scientific #12091021) to the following final concentrations: 180 U/ml zymolase, 20 mM dithiotreitol and 0.2 mg/ml RNase A. The cells were incubated in this solution for 2–3 hr at 35 °C, followed by 10 min at 99 °C to inactivate zymolase. After cool-down to room temperature, Qiagen's QIAprep Spin Miniprep Kit was used according to the manufacturer's instructions but omitting the first step (Qiagen #27104).

The plasmid region containing the gRNA target sequence and the barcode was amplified using the KAPA HiFi HotStart ReadyMix (KAPA/Roche KK2602) supplemented with EvaGreen (Biotium #31000) using the following thermocycling protocol: 45 s at 98 °C, 22 x (15 s at 98 °C, 30 s at 55 °C, 30 s at 72 °C), 1 min at 72 °C. This was performed on an Agilent AriaMx Real-time PCR system in order to be able to monitor and stop the amplification before it started to plateau, which was at 22 cycles. The forward (OS140-149) and reverse (OS125, PAGE-purified) primers included overhangs that mimic the product of the Illumina Nextera transposase as described above. The forward primer also included a stagger sequence of 9–18 nucleotides to introduce diversity in the sequencing library (*Joung et al., 2017*). A second round of PCR (6 cycles) to extend the Nextera adapters was performed using the KAPA HiFi HotStart ReadyMix (KAPA/Roche KK2602) again, together with unique dual indexes (IDT for Illumina DNA/RNA UD Indexes, Integrated DNA Technologies). Concentrations of PCR products were determined using Qubit reagents (dsDNA HS Assay Kit, Thermo Fisher Scientific Q32851) but in 96-well format with fluorometric readings at 485/20 nm excitation and 528/20 nm emission performed on a BioTek Synergy2 plate reader using black opaque flat-bottom 96-well plates (Costar #3915). We then combined roughly equal amounts of DNA from samples of cells sorted for high or low GFP levels and double that amount for the control samples that were sorted without GFP gates. The pooled sample was run on a 2% agarose gel and gel-extracted (Zymo research D4007). The amplicon library was quantified by Qubit (dsDNA HS Assay Kit, Thermo Fisher Scientific Q32851). To increase diversity, PhiX Sequencing Control v3 (Illumina #FC-110–3001) was added and the final library was sequenced on an Illumina NovaSeq 6000 using an S1 flow cell with paired-end 100-base pair reads.

Due to underrepresentation of some of the sorted Tdh3-GFP sub-libraries in the final library mentioned above, we pooled these samples separately again, and gel-purified and quantified them as described above, followed by addition of PhiX Sequencing Control v3 (Illumina #FC-110–3001) and sequencing on an Illumina NextSeq 500 in HighOutput mode with paired-end 80- and 70-base pair reads. Read counts were then downsampled using Seqtk (version 1.3, RRID:SCR_018927, https://github.com/lh3/seqtk) to get comparable read numbers to the other GFP strains.

## Fitness screen analysis

Fitness screen analyses were performed with custom R scripts run in RStudio (R version 4.0.3, RStudio version 1.3). For the fitness screen data shown in *Figure 2C* (ES gRNA set only, two replicates), we first extracted guide sequences from sequencing reads and tallied the read counts per guide. To determine which gRNAs change in abundance during prolonged culturing of base edited yeast, we used read counts per gRNA as a quantitative measure. We normalized the read counts across samples so that the total read counts of the control gRNAs remain constant. We then calculated for each gRNA the $\log_2$ fold change (log2FC) of the number of reads at each time point versus to the "pre" time point

sampled before induction of the base editor with galactose. The reported log2FC values are averages across the two replicate experiments. To obtain gene-level effects, we used the largest absolute log2FC among all gRNAs targeting the same gene. The fitness screen data shown in *Figure 5E and F* (EP, NP, and NS gRNA sets, four replicates) was processed as described above but with normalization for total read counts across samples because not all four replicates included the control gRNAs.

## Protein abundance screen analysis

Protein abundance screen analyses were performed with custom R scripts run in RStudio (R version 4.0.3, RStudio version 1.3). First, for each sample, the 20-nucleotide guide and the 20-nucleotide barcode sequences were extracted from paired-end read pairs. Next, for all barcodes paired with guides perfectly matching the library, we performed error correction to account for potential sequencing errors that would otherwise lead to inflated barcode counts. For this, read counts of barcodes with a difference of less than five nucleotides were assigned the same barcode sequence. We then removed barcodes that did not have at least two reads and gRNAs that did not have at least two barcodes in at least four of the control samples per GFP strain. We also excluded replicates with very low read counts (Eno2_A, Fas1_A, Fas1_F, Fas2_A, Fas2_D, Htb2_A). To determine which gRNAs are differentially abundant among cells with high vs low GFP levels, we used the number of unique barcodes per gRNA as a quantitative measure which we normalized across samples by total unique barcode count per sample. We then applied a generalized linear model (glm) with Poisson-distributed errors and log as a link function. The glm function outputs for each gRNA the ln('high GFP'/ 'low GFP') and a p-value. We converted the ln fold change to $\log_2$ fold change (log2FC). In cases where for a gRNA no barcodes were detected in any of the high or low GFP replicates, the glm function assigns an estimate of 20 or –20, respectively, which we replaced by 3 or –3, respectively. To adjust p-values for multiple testing and control the false discovery rate (FDR), we used the qvalue R package version 2.22.0 (*Storey, 2002*). Unless otherwise noted, we consider 'significant' a q-value (FDR) <0.05.

To obtain gene-level log2FC values, we used the largest absolute fold change estimate of all gRNAs targeting a gene. For gene-level q-values, we first combined the p-values of all gRNAs targeting the same gene using Fisher's method (R package poolr, version 0.8–2) and then corrected these gene-level p-values for multiple testing again with the qvalue R package. We compared this combined q-value to the q-value of the gRNA with the largest absolute fold change per gene and then took the lower of the two q-values as a final per-gene q-value. This approach raises the number of genes with significant effect (FDR <0.05) on protein abundance from 622 to 710, compared to considering just genes targeted by significant gRNAs.

## Cytoscape network representations

For network visualizations we used Cytoscape (version 3.8.2) (*Shannon et al., 2003*). For the network representing the overall results from the screen (*Figure 3B*), the eleven proteins were defined as source nodes and the significant gene perturbations as target nodes. A diverging red-blue color gradient was chosen to reflect the log fold changes caused by the perturbation. Note that for perturbations that affect multiple proteins, the selected color depends on the order the interactions are listed in the input. The size of the target nodes correlates with the number of source nodes (proteins) it connects to. The arrangement of nodes was obtained by the yFiles Organic Layout or the Edge-weighted Spring Embedded Layout and further adjusted manually.

For the small protein-protein-interaction network of the broad-effect gene perturbations (*Figure 5C*), the Cytoscape StringApp was used (*Doncheva et al., 2018*; *Szklarczyk et al., 2020*). The settings for STRING were as follows: the species was set to *Saccharomyces cerevisiae* and the confidence cutoff was set to 0.4. All view defaults were unselected. The type of chart to draw was set to pie chart, with two terms to chart and an overlap cutoff of 0.5. The arrangement of nodes was obtained by applying the yFiles Organic Layout and further adjusted manually.

## Gene Ontology (GO) enrichment analysis

Enrichment analysis for functional annotations was performed in R using the STRINGdb package version 2.2.2 (*Szklarczyk et al., 2020*). This package computes the enrichment using a hypergeometric test and adjusts p-values for multiple testing using the Benjamini-Hochberg method. For the GO enrichment analysis among broad and specific regulators, the background was defined as all

genes represented by our gRNA library (NS, ES, EP subsets). For the GO enrichment analysis among sets of significantly changed proteins in the proteomics data, the background was defined as the union of proteins quantified across all samples used for the enrichment analysis.

### Validation experiments (*RAS2*, *IRA2*, *SSY5*, *POP1*, *SIT4*)

For validation of the findings from the protein screen, we generated individual mutants based on the BY4741 strain. We first cloned individual gRNA plasmids for relevant gRNAs targeting *RAS2*, *IRA2*, *SSY5*, *POP1* and *SIT4* as described above in the 'Plasmids' section. These plasmids were then individually transformed into the BY4741 yeast strain previously transformed with pGal-BE3 (described in the section 'Base editor characterization with individual gRNAs') using the same high-efficiency lithium acetate-based protocol with minor modifications (*Gietz and Schiestl, 2007a*). From the successful transformants growing on selective plates (SC -Leu -Ura) we then inoculated an overnight preculture in selective glucose media (SC -Leu -Ura) followed by a 24 hr growth period in selective galactose media (SC/Gal -Leu -Ura) to induce base editor expression. Subsequently, cells were grown in non-selective liquid YPD media for 8 hr to promote plasmid loss, followed by growth on YPD plates for up to 5 days. Note that colonies on some plates grew very slowly because several of the targeted base edits affect cell fitness. From each plate, we analyzed 4–8 individual colonies to see whether they acquired the expected base edits. First, a small amount of each colony was transferred into a 96-well PCR plate containing 50 µl ultra-pure $H_2O$ (Invitrogen/Thermo Fisher Scientific #10977015). Of the resulting cell solution, 25 µl was transferred into a 96-deep-well plate containing 500 µl YPD. The remaining 25 µl were boiled at 99 °C for 5 min to lyse the cells. Then 2 µl of the heat-treated cells were used to set up 50 µl PCR reactions with TaKaRa Ex Taq DNA Polymerase (TaKaRa #RR001A) and specific primer pairs designed to amplify the genomic region targeted by each gRNA. The following thermal cycling protocol was used: 1 min at 94 °C, 35 x (30 s at 94 °C, 30 s at 50 °C, 1 min at 72 °C), 3 min at 72 °C. PCR products were sent for Sanger sequencing. Three independent mutant strains harboring the expected base edits were then selected and the corresponding cultures in YPD that were kept at room temperature were converted into cryo stocks and stored at –80 °C until use.

### LC-MS proteomics data acquisition

For proteomics analyses, three independent mutant strains per mutation of interest (see above) were streaked out on YPD plates and incubated at 30 °C for several days. Overnight cultures were then inoculated from a single colony of each mutant strain in SC media. The next day, 3 ml cultures were set up at 2.5e6 cells/ml ($OD_{600}$=0.05) and grown for several hours until they reached a density of 2.5e7 cells/ml ($OD_{600}$=0.5). Of each culture, 2 ml were harvested by centrifugation for 1 min at 5000 g at 4 °C. Supernatant was removed and cell pellets stored at –80 °C. For cell lysis, 200 µl freshly prepared lysis buffer (8 M urea, 75 mM NaCl, 50 mM Tris-HCl at pH 8) was added to each cell pellet. The resulting solution was transferred to 2 ml screw-cap tubes containing a 100 µl volume of glass beads (500 µm, Sigma G8772). Cells were lysed by 3x5 min bead beating at 2400 rpm (40 Hz) on a BioSpec Mini-BeadBeater-96, with 1–2 min cooling on ice between cycles. Samples were then centrifuged for 5 min at full speed (at room temperature to avoid precipitation of urea) and cleared lysates transferred to new tubes. Protein concentrations were determined with a colorimetric bicinchoninic acid (BCA) assay (Pierce / Thermo Scientific #23225). Absorbance at 562 nm was measured on a BioTek Synergy2 plate reader which indicated protein concentrations of 1–2 mg/ml.

For each sample, 25–50 µg of protein was further processed as follows. Protein disulfide bonds were reduced by the addition of 5 mM Tris (2-carboxyethyl)phosphine (TCEP) and incubation at room temperature for 30 min. Next, the free cysteine residues were alkylated by the addition of 10 mM iodoacetamide and incubation at room temperature for 30 min in the dark. Subsequently, samples were diluted with 100 mM Tris-HCl at pH 8 to reach a urea concentration of less than 2 M. For the protein digest, 0.2 µg LysC (Wako #125–05061) and 0.8 µg trypsin (Pierce / Thermo Scientific, #90057) were added and samples were incubated at 37 °C overnight. The digest was quenched by adding formic acid to a final concentration of 5% (v/v). Finally, samples were desalted using C18 tips (Pierce / Thermo Scientific #87784), dried under vacuum and reconstituted in 5% formic acid (v/v).

For liquid chromatography-coupled tandem mass spectrometric (LC-MS/MS) analysis, desalted peptides were fractionated online by nano-flow reversed phase liquid chromatography using a Dionex UltiMate 3000 UHPLC system (Thermo Fisher Scientific). The column consisted of a 25 cm fused

silica capillary with 75 μm inner diameter that was packed in-house (*Jami-Alahmadi et al., 2021*) with ReproSil-Pur C18 particles (particle size 1.9 μm; pore size 120 Å; Dr. Maisch). The linear 140 min water–acetonitrile gradient from 5% to 35% acetonitrile was delivered at a flow rate of 300 nl/min. Mass spectrometric analysis was performed on an Orbitrap Fusion Lumos Tribrid mass spectrometer (Thermo Fisher Scientific) run in Data-Dependent Acquisition (DDA) mode and with an inclusion list containing a total of six peptides from Tdh1 and Tdh2 to ensure their consistent quantification (IDVA-VDSTGVFK(2+), LISWYDNEYGYSAR(2+), VVDLIEYVAK(2+), HIIVDGHK(2+), DPANLPWASLNIDIAI DSTGVFK(2+), GVLGYTEDAVVSSDFLGDSNSSIFDAAAGIQLSPK(3+)). MS1 spectra were acquired at a resolution of 120,000 (FWHM at 400 m/z) over a range of 400–1600 m/z with a maximum injection time of 100ms. MS1 acquisition was followed by the acquisition of MS2 spectra from precursors falling into the inclusion list windows (monoisotopic m/z+/-25 ppm). Precursors were isolated with a window of 1.6 m/z and fragmentation was obtained by higher-energy C-trap dissociation (HCD) at a fixed colli-sion energy of 35%. MS2 spectra were acquired at a resolution of 30,000, a maximum injection time of 54ms and dynamic exclusion for 5 s. The remainder of the 3 s cycle time was used to acquire MS2 spectra for other precursors at a resolution of 15,000 and an exclusion window of 25 s but otherwise identical parameters as for the inclusion list precursors.

## LC-MS proteomics data analysis

Peptide identification and quantification was performed with MaxQuant version 1.6.17.0 (*Cox and Mann, 2008*). The search was performed against a *S. cerevisiae* S288C protein database containing 5917 protein sequences (version R64-2-1, *Saccharomyces* Genome Database (SGD)). Only fully LysC- and trypsin-digested peptides with up to one missed cleavage were allowed. LysC was specified to cleave after lysine even if followed by proline and trypsin was specified to cleave after lysine and argi-nine if not followed by proline. Carbamidomethylation of cysteines was defined as a fixed modification and methionine oxidation and N-terminal acetylation as a variable modification. Peptide-spectrum-match (PSM) and protein-level false discovery rates (FDRs) were set to 0.01. Further analyses were performed with custom R scripts run in RStudio (R version 4.0.3, RStudio version 1.3). From the MaxQuant output (evidence.txt), all non-unique peptides were removed before further processing. Relative quantification was then performed in R using the artMS package version 1.8.3 (http://artms. org), which itself is based on the MSstats package (*Choi et al., 2014*). Briefly, $\log_2$-transformed data was quantile normalized and summarized by Tukey's median polish. Missing values were imputed and p-values were adjusted for multiple testing by the Benjamini-Hochberg method. Enrichment analysis for functional annotations among sets of significantly changed proteins in different mutant strains was performed as described above with the union of proteins quantified across all samples used for the enrichment analysis as background.

## Testing effect of sequence context on editing efficiency

To test whether the sequence context in the targeting region of the gRNA has an effect on editing efficiency, we made use of the data from the fitness screen, considering only gRNAs introducing stop codons into essential genes (ES subset) and the control gRNAs without a C in the target region. We considered the 20-nucleotide gRNA targeting sequence flanked on either side by an additional five nucleotides for a total of 30 nucleotides per gRNA. For each of the 5928 gRNAs, we then extracted all possible position-specific sequence features up to 4 nucleotides in length, resulting in a total of over 8000 sequence features across all gRNAs (*Rahman and Rahman, 2017*). The gRNAs were split into a training set of 4928 gRNAs and a test set of 1000 gRNAs. An ordinary and a logistic lasso regression model was then obtained from the training set. For the logistic lasso model, we defined gRNAs with a $\log_2$ fold change <–0.5 as dropping out. Evaluated on the test set, the lasso-predicted and observed effects show a Pearson's correlation coefficient ($R^2$) of 0.37 and 0.35 for the ordinary and the logistic model, respectively.

## Comparison to eQTL and pQTL hotspots in natural yeast isolates

To date, the most comprehensive picture of genetic architecture underlying gene expression regu-lation in yeast has been achieved by linkage mapping in crosses of divergent yeast strains (*Albert et al., 2018*). A remarkable finding is that a majority of genetic loci affecting mRNA and protein expression (eQTLs and pQTLs, respectively) cluster in regulatory hotspots. However, these regulatory

hotspots can be large genomic regions encompassing up to dozens of genes each, making it challenging to pinpoint the causal gene and, hence, to understand their mechanism of action. We set out to use the results from the base editor screens to identify candidate causal genes that could underlie the observed eQTL and pQTL hotspots in segregants of the *S. cerevisiae* strains BY S288C and RM11-1a (*Albert et al., 2018*; *Albert et al., 2014*). Among the genes located in the eQTL and pQTL hotspot intervals, we identified those that affect at least three of the eleven proteins studied here. The resulting list of gene candidates was narrowed down to those that contain at least one variant between BY and RM with a predicted medium or strong effect (i.e., missense mutation, indel, or premature stop codon). We were able to identify for 24 eQTL and pQTL hotspots a total of 47 candidate causal genes, each affecting three or more of the eleven proteins and harboring at least one parental variant with a predicted medium to strong effect (*Supplementary file 5*). For only three of these 24 regulatory hotspots the causal gene is known and our screen identified two of them, *IRA2* and *OLE1*, as candidate causal genes (*Lutz et al., 2019*; *Smith and Kruglyak, 2008*).

## Acknowledgements

We thank Dr. James Boocock and Dr. Frank Albert for helpful scientific discussions and Dr. Oliver F Brandenberg for feedback on the manuscript. We also thank Dr. Yasaman Jami-Alahmadi and Dr. James A Wohlschlegel from the UCLA Proteomics Facility for proteomics measurements and Dr. Xinmin Li and his team at the UCLA Technology Center for Genomics & Bioinformatics for sequencing services. This work was supported by fellowships and grants from the Human Frontier Science Program (to OTS, LT000737/2016 L), the Swiss National Science Foundation (to OTS, P2EZP3_165280), the Howard Hughes Medical Institute (to LK) and the National Institutes of Health (to LK, R01GM102308). The authors declare no competing financial interests.

## Additional information

### Funding

| Funder | Grant reference number | Author |
|---|---|---|
| Human Frontier Science Program | LT000737/2016-L | Olga T Schubert |
| Swiss National Science Foundation | P2EZP3_165280 | Olga T Schubert |
| National Institutes of Health | R01GM102308 | Leonid Kruglyak |
| Howard Hughes Medical Institute | Investigator | Leonid Kruglyak |

The funders had no role in study design, data collection and interpretation, or the decision to submit the work for publication.

### Author contributions

Olga T Schubert, Conceptualization, Data curation, Formal analysis, Funding acquisition, Validation, Investigation, Visualization, Methodology, Writing – original draft, Project administration, Writing – review and editing; Joshua S Bloom, Conceptualization, Formal analysis, Methodology, Writing – review and editing; Meru J Sadhu, Conceptualization, Methodology, Writing – review and editing; Leonid Kruglyak, Conceptualization, Supervision, Funding acquisition, Writing – review and editing

### Author ORCIDs

Olga T Schubert http://orcid.org/0000-0002-2613-0714
Joshua S Bloom http://orcid.org/0000-0002-7241-1648
Leonid Kruglyak http://orcid.org/0000-0002-8065-3057

### Decision letter and Author response

Decision letter https://doi.org/10.7554/eLife.79525.sa1

Author response https://doi.org/10.7554/eLife.79525.sa2

## Additional files

### Supplementary files
• Supplementary file 1. gRNAs used in this study.

• Supplementary file 2. Yeast strains used in this study.

• Supplementary file 3. Plasmids used in this study.

• Supplementary file 4. Primers used in this study.

• Supplementary file 5. Candidate causal genes for eQTL and pQTL hotspots. The table lists candidate causal genes for previously described eQTL (*Albert et al., 2018*) and pQTL (*Albert et al., 2014*) hotspots identified in segregants of a cross between the *S. cerevisiae* strains BY S288C and RM11-1a. The candidate genes each affect at least three of our eleven proteins and contain at least one variant between BY and RM with a predicted medium or strong effect (i.e., missense mutation, indel, or premature stop codon). Note that the pQTLs were mapped for 160 proteins, five of which overlap with the eleven proteins studied here (Tdh1, Tdh3, Rpl9A, Ssa1, Yhb1); the eQTLs were mapped for transcripts of thousands of genes, including the genes encoding the eleven proteins studied here.

• MDAR checklist

### Data availability
The DNA sequencing read data has been deposited in the NCBI Sequence Read Archive under BioProject ID PRJNA732764. The mass spectrometry proteomics data has been deposited in the ProteomeXchange Consortium partner repository PRIDE with the dataset identifier PXD029363. All custom code is available on GitHub without restrictions: https://github.com/OlgaTSchubert/Protein-Genetics, (copy archived at swh:1:rev:bab9abdb34f8ab883e5881f9d9a24abfdd1e0a6c).

The following datasets were generated:

| Author(s) | Year | Dataset title | Dataset URL | Database and Identifier |
|---|---|---|---|---|
| Schubert OT, Bloom JS, Sadhu MJ, Kruglyak L | 2022 | Base editor screen for protein abundance | https://www.ncbi.nlm.nih.gov/bioproject/PRJNA732764 | NCBI BioProject, PRJNA732764 |
| Schubert OT, Bloom JS, Sadhu MJ, Kruglyak L | 2022 | Proteomics validation | https://doi.org/10.6019/PXD029363 | ProteomeXchange, 10.6019/PXD029363 |

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
