## [Editor Report]

This paper describes a novel CRISPR-based screening method that allows probing interactions between a large set of specific mutations and the abundance of specific proteins, and, more generally, investigate the spectrum of effects that (point) mutations can have on protein abundance. This novel technique complements existing strategies for measuring effects of genetic perturbations on transcript levels, which is important as for some genes mRNA and protein levels may not correlate well. The ability to measure proteins directly therefore promises to help close a gap in our understanding of the links between genotype and phenotype, and the strategy is broadly applicable beyond the current study.

---

## [Decision Letter]

**Decision letter after peer review:**

Thank you for submitting your article "Genome-wide base editor screen identifies regulators of protein abundance in yeast" for consideration by *eLife*. Your article has been reviewed by 3 peer reviewers, including Kevin J Verstrepen as the Reviewing Editor and Reviewer #1, and the evaluation has been overseen by Naama Barkai as the Senior Editor. The following individual involved in review of your submission has agreed to reveal their identity: Sibylle Vonesch (Reviewer #2).

We really like the study and believe that it merits publication in *eLife*, with two main areas that can help improve it even further.

Firstly, data and explanation on the setup and control experiments (as suggested in detail by reviewers 2 and 3) should be added.

Second, and related, it would benefit the reader to choose one clear strategy to present the study (i.e., a methods paper or a paper that describes a few specific novel regulatory interactions). It seems clear to us that you intend it more as a "methods" paper, and we agree that this is the best way to present the results. However, in its current form, the paper still floats somewhat in between. A few textual changes (in abstract, methods and conclusions) may be sufficient to improve the flow; eg by very clearly putting the parts that really concern the development and validation of the method first in every section, and better separating these from the results that come out of the validation experiments.

We realize that the second suggestion may imply that the biological results become a bit hidden, but perhaps these can then take center stage in a follow-up study that addresses some of these biological questions (eg general vs specific regulators of protein abundance, specific novel regulatory interactions for specific proteins or processes) more exhaustively?

We include the individual reports from the three reviewers to this message to provide more details and also share the other suggestions that you might want to take into account.

*Reviewer #1 (Recommendations for the authors):*

Is it fair to call the effects of random mutations on the abundance of (some) proteins "regulatory"? To me, the concept of regulation is perhaps a bit more narrow and precise, involving a mechanism that helps organisms adapt to specific signals. In that view, the fact that a random point mutation influences the abundance of a random protein falls short of being a regulatory relationship; I see this as an "interaction" or "relationship" (as it is also often referred to in the body of the text). For example, when you perturb a heme transporter, Yhb1 might respond to compensate, but I do not think it is fair to call that a regulatory relationship without showing that the mutated proteins play a role in sensing and/or signaling. This is different for perturbations in Hap1, as that is really a (transcriptional) regulator, or for the mutations in Ssy1-Ptr3-Ssy5, which are also involved in regulation.

One of the interesting observations is that some reporter proteins, like Rpl9, are affected by a much broader set of mutations compared to others, like Tdh1. It could be interesting to speculate why that is. An obvious hypothesis would be that genes that are "hubs" and that interact with many other genes are influenced by a larger number of mutations. One way to check this would be to correlate the results with previous results from genome-wide interaction studies, such as the ones carried out by Charlie Boone's team, and/or by mapping the reporters on existing interaction networks and somehow quantifying the size of their immediate/neighboring network…

Results: Perhaps better to report R2 rather than R values for correlations between the screen's result and the confirmation with single gRNAs?

*Reviewer #2 (Recommendations for the authors):*

The introduction is geared towards the point that mRNA levels are not always predictive for protein levels and therefore looking at proteins directly is relevant. But this aspect is not really much discussed further either in the introduction or in the results/discussion. For me this would be an interesting point, how much or what do we gain in knowledge by directly measuring protein that we did not know/that was different when looking at transcript levels. The introduction is in general a bit short and could be extended. It would be nice to compare results with those obtained for transcriptomes measured in KO strains (https://doi.org/10.1016/j.cell.2014.02.054), see also comment in public review.

I would have liked to see more of a comparison between the effects of the missense mutations compared to the stop codons as this really emphasizes the advantage of using a base editor (vs just screening a KO library).

Mediator/Cdk8 and protein abundance: The effect seems to be limited to Htb2 and Tdh1/2, so this should be formulated accordingly. Any speculation why it would be confined to these and not others? Does it have to do with how strongly proteins are regulated at the transcriptional vs translational level? (for example, is there a correlation with how well mRNA levels explain protein level for these proteins?)

Figure 3D: Are there any explanations for the discrepancy of Tdh1 levels measured with the reporters and MS?

It could be interesting to look at the similarity of deletions based on perturbation profiles of the eleven proteins. Are there common signatures apart from ribosome biogenesis/tRNA metabolism and transcription? For example, similar effects for complexes or pathways? This is shown individually for some complexes (SDS, mediator) but a more global analysis would be interesting.

The effect of SDS seems quite global (affecting all/most of the proteins to some extent, vs the transcription factor Stp1/2 does not, Figure 2B). Could this point at a more global role of SDS or could it be a general consequence of cells perceiving amino acid limited conditions? Stp1 is a direct regulator of Yhb1, but has a lower effect on Yhb1 levels than the SPS complex. Is it possible that SDS also interacts directly or via another input to Yhb1?

Editing in Figure S1 (characterization of BE3 in yeast) was quite long (44h) vs shorter (24h) in the fitness and protein screen. Do the authors have any information on how that would have impacted editing efficiency and purity?

Table S7: Which proteins were the eQTLs and pQTLs originally mapped for? The same ones as are measured with the translational fusions?

Figure S6: Would this plot not make more sense split by stop codon and missense? It might be dominated by the stop codons and mask the pattern for missense alleles.

Figure 3A: indicating percentages would help, hard to interpret

Addgene: plasmids not yet available

*Reviewer #3 (Recommendations for the authors):*

1. It could be worth including an additional figure as the new figure 1 that summarizes the validation experiments for the CRISPR base editor. This new technology alone seems like an important achievement, and I think many readers will be interested in applying the system to their own research. So, the results of these validation experiments could be of broad interest.

2. I had some trouble in the section of the paper that talks about regulation at the translational vs. transcriptional levels. Specifically, at first, I did not understand why the authors felt the following result was remarkable: "Remarkably, 21 (72%) of these 29 genes have roles in protein translation-more specifically, in ribosome biogenesis and tRNA metabolism (FDR < 8.0e-4, Figure 3C)." I also did not understand why this was being contrasted with transcription in the subsequent sentence: "In contrast, perturbations that affect the abundance of only one or two of the eleven proteins mostly occur in genes with roles in transcription (e.g., GO:0006351, 98 genes, 47 expected, FDR < 1.3e-5)." Upon re-reading the introduction, I remembered that the authors claim their study is one of the few that look at protein levels rather than transcript levels. So, I understood why they made the comparison. My first suggestion is that the authors should remind readers of the relevance of this comparison. But as soon as I thought I understood, I again became confused. Isn't it obvious that the genes affecting protein levels will be the genes involved in translation rather than transcription? So, I'm still confused about why this result is remarkable, rather than just expected. This problem is deeper than the use of the word, "remarkable" because much of the intro and discussion focus on this different in translation vs. transcription. Perhaps the authors should elaborate on why they expected more proteins involved in transcription to influence protein abundance.

3. Relatedly, this point in the discussion seems super cool: "Genes affecting the abundance of many proteins simultaneously (i.e., hubs of the protein regulatory network) were often involved in post-transcriptional protein biosynthetic processes. In contrast, perturbations that affected only specific proteins were enriched in transcriptional regulation." When I first read this, I was not sure where this was covered in the paper. In fact, its discussed in the two sentences I just quoted in major comment 2. Clearly, those sentences were lost on me! Maybe consider adding more here, perhaps a figure? Or more statistical analysis? The authors describe the overrepresentation of transcriptional genes that affect 1 or 2 of the 11 targets. Is there a corresponding underrepresentation of translational genes?

4. Another thing I wondered about this comparison between studies of protein abundance vs. transcript abundance is how does the power of these types of studies compare. Here the authors focus on 11 target proteins. Is that similar to transcript studies, or do those studies have power to focus on more targets? Why have studies of protein abundance previously been limited? It seems like this base editor could be used for either type of study, so I'm not sure how it helps specifically with protein abundance studies.

5. I also wondered if the conclusions about the connectedness of the protein regulatory network differ from those about the connectedness of transcriptional networks. Is one more connected than the other?

6. I found figure 3E to be jam packed, and I also use pooled competitions to study relative fitness of many yeast strains, so if I was confused I imagine others will be totally lost. It might be good to add an additional panel showing some examples of an oligo that declines in frequency vs, one that does not? This helps with my second issue as well, which is, how do you know things in the last bin (8+ targets) didn't just start at lower frequency, so more likely to be lost due to stochastic process? I see the number of things in each bin drops off with the number of targets affected, so the 6, 7, and 8+ bins have the fewest things in them. Maybe this makes them more sensitive to stochastic effects? It would be nice to see that some of the things in the 8+ bin start out at the same frequency as some of the things in the 1 or 2 bin, and yet are still lost. You could do this in the same panel that gives the reader a better understanding of how fitness is measured. Or maybe my concern could be alleviated in a simple sentence that explains the range of starting frequencies in the 6, 7, and 8+ bin was equivalent to those in other bins. I leave this up to the authors.

---

## [Author Response]

We really like the study and believe that it merits publication in eLife, with two main areas that can help improve it even further.

We really appreciate the positive reception of our work and the constructive comments for improvement.

Firstly, data and explanation on the setup and control experiments (as suggested in detail by reviewers 2 and 3) should be added.

We expanded the first part of the manuscript with information that was previously in the supplementary materials and added additional information as requested by the reviewers (see specific responses below).

Second, and related, it would benefit the reader to choose one clear strategy to present the study (i.e., a methods paper or a paper that describes a few specific novel regulatory interactions). It seems clear to us that you intend it more as a "methods" paper, and we agree that this is the best way to present the results. However, in its current form, the paper still floats somewhat in between. A few textual changes (in abstract, methods and conclusions) may be sufficient to improve the flow; eg by very clearly putting the parts that really concern the development and validation of the method first in every section, and better separating these from the results that come out of the validation experiments.

We modified the manuscript as suggested.

Reviewer #1 (Recommendations for the authors):Is it fair to call the effects of random mutations on the abundance of (some) proteins "regulatory"? To me, the concept of regulation is perhaps a bit more narrow and precise, involving a mechanism that helps organisms adapt to specific signals. In that view, the fact that a random point mutation influences the abundance of a random protein falls short of being a regulatory relationship; I see this as an "interaction" or "relationship" (as it is also often referred to in the body of the text). For example, when you perturb a heme transporter, Yhb1 might respond to compensate, but I do not think it is fair to call that a regulatory relationship without showing that the mutated proteins play a role in sensing and/or signaling. This is different for perturbations in Hap1, as that is really a (transcriptional) regulator, or for the mutations in Ssy1-Ptr3-Ssy5, which are also involved in regulation.

While we agree in principle, we decided in the interests of brevity to retain the term “regulator” for lack of a better term. We added the following statement to the manuscript to clarify: “Some of these genes may be direct regulators of the tested proteins, while others may affect protein abundances more indirectly. For simplicity and for lack of a better term, we will, in the following, refer to these genes as regulators without implying a direct mechanistic link.”

One of the interesting observations is that some reporter proteins, like Rpl9, are affected by a much broader set of mutations compared to others, like Tdh1. It could be interesting to speculate why that is. An obvious hypothesis would be that genes that are "hubs" and that interact with many other genes are influenced by a larger number of mutations. One way to check this would be to correlate the results with previous results from genome-wide interaction studies, such as the ones carried out by Charlie Boone's team, and/or by mapping the reporters on existing interaction networks and somehow quantifying the size of their immediate/neighboring network…

We thank the reviewer for this suggestion. We compared our results with the number of genetic interactions determined from double-knock out strains (Costanzo et al., Science 2018) and indeed found that among the genes encoding our eleven proteins, RPL9A has the most genetic interactions, while TDH1 has the fewest. Across all eleven proteins, the correlation is not significant (see Author response image 1 ), and more proteins would need to be analyzed for a statistically meaningful comparison.

**Author response image 1. sa2fig1:** 

Results: Perhaps better to report R2 rather than R values for correlations between the screen's result and the confirmation with single gRNAs?

We changed all instances of Pearson’s R to R^2^ in the main text of the manuscript.

Reviewer #2 (Recommendations for the authors):The introduction is geared towards the point that mRNA levels are not always predictive for protein levels and therefore looking at proteins directly is relevant. But this aspect is not really much discussed further either in the introduction or in the results/discussion. For me this would be an interesting point, how much or what do we gain in knowledge by directly measuring protein that we did not know/that was different when looking at transcript levels. The introduction is in general a bit short and could be extended. It would be nice to compare results with those obtained for transcriptomes measured in KO strains (https://doi.org/10.1016/j.cell.2014.02.054), see also comment in public review.

We agree with the reviewer that it would be very interesting to compare the effect of perturbations on mRNA vs protein levels. We have compared our protein-level data to mRNA-level data from Kemmeren and colleagues (Kemmeren et al., Cell 2014), and we find very good agreement between the effects of gene perturbations on mRNA and protein levels when considering only genes with q < 0.05 and Log2FC > 0.5 in both studies (Pearson’s R = 0.79, p < 5.3e-15).

Gene perturbations with effects detected only on mRNA but not protein levels are enriched in genes with a role in “chromatin organization” (FDR = 0.01; as a background for the analysis, only the 1098 genes covered in both studies were considered). This suggests that perturbations of genes involved in chromatin organization tend to affect mRNA levels but are then buffered and do not lead to altered protein levels. There was no enrichment of functional annotations among gene perturbations with effects on protein levels but not mRNA levels. We did not initially include these results in the manuscript because there are some limitations to the conclusions that can be drawn from these comparisons, including that our study has a relatively high number of false negatives, and that the genes perturbed in the Kemmeren et al. study were selected to play a role in gene regulation, meaning that differences in mRNA-vs-protein effects of perturbations are limited to this function, and other gene functions cannot be assessed. We now included the figure in the manuscript (Figure 3-figure supplement 2) and discuss the limitations in the Discussion section.

I would have liked to see more of a comparison between the effects of the missense mutations compared to the stop codons as this really emphasizes the advantage of using a base editor (vs just screening a KO library).

Our main gRNA library that was used for the fitness and protein screens contained—besides gRNAs introducing stop codons—only gRNAs targeting highly conserved amino acid residues (absolute PROVEAN score > 5). The targeted missense mutations are therefore skewed towards loss-of-function phenotypes similar to the stop codons. Hence, the data from the fitness and protein screens are not well suited for such an analysis. However, for the CAN1 screen we used a gRNA library that targeted all possible missense mutations and is therefore unbiased. We added the figure below to the manuscript that shows that some missense mutations have effects as strong as the premature stop codons while others have smaller effects or no effect at all (Figure 2A).

Mediator/Cdk8 and protein abundance: The effect seems to be limited to Htb2 and Tdh1/2, so this should be formulated accordingly. Any speculation why it would be confined to these and not others? Does it have to do with how strongly proteins are regulated at the transcriptional vs translational level? (for example, is there a correlation with how well mRNA levels explain protein level for these proteins?)

We clarified in the manuscript that the effect of the perturbations of the discussed mediator subunits is mostly on these three proteins. As mentioned in the section on the GAPDH isoenzymes, the regulatory module Cdk8 is involved in carbon catabolite repression, and Tdh1/2 are likely regulated by this system, while most of the other proteins are not. In the mRNA dataset by Kemmeren and colleagues, this regulatory pattern in response to perturbations of subunits of the Cdk8 module is not evident.

Figure 3D: Are there any explanations for the discrepancy of Tdh1 levels measured with the reporters and MS?

The proteomics/MS samples for the POP1 mutant were collected at slightly higher OD than the sampling for the cell sorting/GFP. It is known that Tdh1 expression is strongly induced in stationary cells (Delgado et al., Microbiology 2001), and it is possible that in the POP1 mutant cultures collected for proteomics it is already responding to higher cell density.

It could be interesting to look at the similarity of deletions based on perturbation profiles of the eleven proteins. Are there common signatures apart from ribosome biogenesis/tRNA metabolism and transcription? For example, similar effects for complexes or pathways? This is shown individually for some complexes (SDS, mediator) but a more global analysis would be interesting.

We agree with the reviewer that this would be very interesting. However, unfortunately, we found that the number of proteins studied here is too small to do such analyses properly.

The effect of SDS seems quite global (affecting all/most of the proteins to some extent, vs the transcription factor Stp1/2 does not, Figure 2B). Could this point at a more global role of SDS or could it be a general consequence of cells perceiving amino acid limited conditions? Stp1 is a direct regulator of Yhb1, but has a lower effect on Yhb1 levels than the SPS complex. Is it possible that SDS also interacts directly or via another input to Yhb1?

We agree with the reviewer that SPS vs Stp1 effects on protein levels are interesting because they do not follow the expectation that Stp1 is the direct effector of SPS. The suggested hypotheses would need to be tested in follow-up experiments beyond the scope of this manuscript.

Editing in Figure S1 (characterization of BE3 in yeast) was quite long (44h) vs shorter (24h) in the fitness and protein screen. Do the authors have any information on how that would have impacted editing efficiency and purity?

Experiments with individual gRNAs targeting genomically inserted GFP show that a majority of cells contains the desired edit within 24 hours of base editing, and we think the first experiment where the base editor was induced for 44 hours is therefore likely to have resulted in similar results to the 24 hours used afterwards for the screens. The purity is likely to be higher, but we do not have the data to show this. We added this information in a supplementary figure and the following sentence in the main text: “To assess base editing efficiency over time, we designed two gRNAs predicted to lead to loss-of-function mutations in GFP. We expressed the base editor and these gRNAs in yeast cells with a genomically integrated GFP gene. Depending on the gRNA, cells lost the GFP signal at different rates; however, for both gRNAs, a majority of cells had lost the GFP signal after 24 hours (Figure 1–figure supplement 3).” We also added a paragraph in the Methods section to describe this experiment.

Table S7: Which proteins were the eQTLs and pQTLs originally mapped for? The same ones as are measured with the translational fusions?

The pQTLs were mapped for 160 proteins (from the same GFP collection); 5 of our eleven proteins overlap with these 160 (Tdh1, Tdh3, Rpl9A, Ssa1, Yhb1). The eQTLs were mapped for thousands of genes, including all our eleven proteins. We added a sentence in the legend of Supplementary File 5 (formerly Table S7) that contains this information.

Figure S6: Would this plot not make more sense split by stop codon and missense? It might be dominated by the stop codons and mask the pattern for missense alleles.

Because the missense mutations introduced by our gRNAs are at highly conserved sites, it is likely that they have a loss-of-function effect similar to premature stop codons. Accordingly, both mutation types show a similar position-dependent trend as shown in the Author response image 2.

Figure 3A: indicating percentages would help, hard to interpret

We modified the figure by changing it from a pie chart to a stacked bar chart and added percentages and more text to improve interpretability (Figure 5A).

Addgene: plasmids not yet available.

The plasmids are ready to be released publicly. We are just waiting for the manuscript to be accepted so that we can add the proper reference to the Addgene entries.

Reviewer #3 (Recommendations for the authors):1. It could be worth including an additional figure as the new figure 1 that summarizes the validation experiments for the CRISPR base editor. This new technology alone seems like an important achievement, and I think many readers will be interested in applying the system to their own research. So, the results of these validation experiments could be of broad interest.

We extended the first part of the manuscript and added new figures (some from the supplementary materials, others are new).

2. I had some trouble in the section of the paper that talks about regulation at the translational vs. transcriptional levels. Specifically, at first, I did not understand why the authors felt the following result was remarkable: "Remarkably, 21 (72%) of these 29 genes have roles in protein translation-more specifically, in ribosome biogenesis and tRNA metabolism (FDR < 8.0e-4, Figure 3C)." I also did not understand why this was being contrasted with transcription in the subsequent sentence: "In contrast, perturbations that affect the abundance of only one or two of the eleven proteins mostly occur in genes with roles in transcription (e.g., GO:0006351, 98 genes, 47 expected, FDR < 1.3e-5)." Upon re-reading the introduction, I remembered that the authors claim their study is one of the few that look at protein levels rather than transcript levels. So, I understood why they made the comparison. My first suggestion is that the authors should remind readers of the relevance of this comparison. But as soon as I thought I understood, I again became confused. Isn't it obvious that the genes affecting protein levels will be the genes involved in translation rather than transcription? So, I'm still confused about why this result is remarkable, rather than just expected. This problem is deeper than the use of the word, "remarkable" because much of the intro and discussion focus on this different in translation vs. transcription. Perhaps the authors should elaborate on why they expected more proteins involved in transcription to influence protein abundance.

We thank the reviewer for pointing out that this paragraph requires more explanation. We expanded it as follows: “Of these 29 genes, 21 (72%) have roles in protein translation—more specifically, in ribosome biogenesis and tRNA metabolism (FDR < 8.0e-4, Figure 5C). In contrast, perturbations that affect the abundance of only one or two of the eleven proteins mostly occur in genes with roles in transcription (e.g., GO:0006351, FDR < 1.3e-5). Protein biosynthesis entails both transcription and translation, and these results suggest that perturbations of translational machinery alter protein abundance broadly, while perturbations of transcriptional machinery can tune the abundance of individual proteins. Thus, genes with post-transcriptional functions are more likely to appear as hubs in protein regulatory networks, whereas genes with transcriptional functions are likely to show fewer connections.”

3. Relatedly, this point in the discussion seems super cool: "Genes affecting the abundance of many proteins simultaneously (i.e., hubs of the protein regulatory network) were often involved in post-transcriptional protein biosynthetic processes. In contrast, perturbations that affected only specific proteins were enriched in transcriptional regulation." When I first read this, I was not sure where this was covered in the paper. In fact, its discussed in the two sentences I just quoted in major comment 2. Clearly, those sentences were lost on me! Maybe consider adding more here, perhaps a figure? Or more statistical analysis? The authors describe the overrepresentation of transcriptional genes that affect 1 or 2 of the 11 targets. Is there a corresponding underrepresentation of translational genes?

Please see our response to Comment #2 above. We did check for underrepresentation of the respective functional classes as suggested by the reviewer but did not find any indication for this.

4. Another thing I wondered about this comparison between studies of protein abundance vs. transcript abundance is how does the power of these types of studies compare. Here the authors focus on 11 target proteins. Is that similar to transcript studies, or do those studies have power to focus on more targets? Why have studies of protein abundance previously been limited? It seems like this base editor could be used for either type of study, so I'm not sure how it helps specifically with protein abundance studies.

With the protein screen described here, we can study only one selected protein at a time, but we can simultaneously test the effects of thousands of genetic perturbations on this protein. This is the opposite of what was done before in studies of genetic effects on mRNA and protein levels that start with a gene knockout and measure the responses of thousands of mRNAs or proteins (for proteins, e.g., Stefely et al., Nature Biotechnology, 2016; for mRNAs, e.g., Kemmeren et al., Cell, 2014). Since our protein screen requires a fluorescent label for the quantification of the protein (and the sorting of the cells), it cannot be easily applied to quantify mRNAs. However, with latest technology developments in single-cell CRISPR-based screening, it is an exciting prospect to do base editor screens and read out the effect of each perturbation on the entire transcriptome by single-cell RNA-sequencing (see, e.g., Przybyla and Gilbert, Nature Reviews Genetics, 2022).

5. I also wondered if the conclusions about the connectedness of the protein regulatory network differ from those about the connectedness of transcriptional networks. Is one more connected than the other?

We agree with the reviewer that this would be interesting to compare but we unfortunately don’t have the appropriate data to directly compare this. Reviewer 1 suggested comparing the connectedness of the protein network to the genetic interaction network by the Boone lab (see Reviewer #1’s Comment #2). Our most connected protein (Rpl9A) is indeed also a hub in the genetic interaction network and our least connected protein (Tdh1) is also least connected in the genetic interaction network. A proper analysis of such trends would require large numbers of proteins.

6. I found figure 3E to be jam packed, and I also use pooled competitions to study relative fitness of many yeast strains, so if I was confused I imagine others will be totally lost. It might be good to add an additional panel showing some examples of an oligo that declines in frequency vs, one that does not?

We thank the reviewer for pointing out that Figure 3E (now 5D/E) was hard to grasp. To address this, we split the figure into two separate subfigures and made the axis labels more explicit. Figure 5E now shows for each gRNA just its fitness effect (vertical axis) and the number of proteins it affects (horizontal axis). Note that the box plot overlay just summarizes the underlying data points.

This helps with my second issue as well, which is, how do you know things in the last bin (8+ targets) didn't just start at lower frequency, so more likely to be lost due to stochastic process? I see the number of things in each bin drops off with the number of targets affected, so the 6, 7, and 8+ bins have the fewest things in them. Maybe this makes them more sensitive to stochastic effects? It would be nice to see that some of the things in the 8+ bin start out at the same frequency as some of the things in the 1 or 2 bin, and yet are still lost. You could do this in the same panel that gives the reader a better understanding of how fitness is measured. Or maybe my concern could be alleviated in a simple sentence that explains the range of starting frequencies in the 6, 7, and 8+ bin was equivalent to those in other bins. I leave this up to the authors.

As shown in Author response image 3, the initial frequencies (read counts) for the gRNAs across all bins are similar. The figure shows the number of reads per gRNA across all bins at the reference time point, i.e., before base editor induction. We added a sentence to the manuscript as suggested: “Note that at the reference time point of the fitness experiment, there was no bias in read counts across the gRNAs affecting different numbers of proteins.”

**Author response image 3. sa2fig3:**